# Planetary boundary layer height by means of lidar and numerical simulations over New Delhi, India

Konstantina Nakoudi[1, 2], Elina Giannakaki[1, 3], Aggeliki Dandou[1], Maria Tombrou[1], Mika Komppula[3]

[1]Department of Environmental Physics and Meteorology, Faculty of Physics, University of Athens, Greece
[2]Alfred Wegener Institute, Helmholtz Centre for Polar and Marine Research, Potsdam, Germany
[3]Finnish Meteorological Institute, Kuopio, Finland

*Correspondence to*: K. Nakoudi (knakoudi@phys.uoa.gr)

**Abstract.** In this work, the height of the Planetary Boundary Layer (PBLH) is investigated over Gual Pahari, New Delhi, for almost a year. To this end, ground-based measurements from a multi-wavelength Raman lidar, were used. The modified Wavelet Covariance Transform (WCT) method was utilized for PBLH retrievals. Results were compared to data from Cloud-Aerosol Lidar and Infrared Pathfinder Satellite Observation (CALIPSO) and the Weather Research and Forecasting (WRF) model. In order to examine the difficulties of PBLH detection from lidar, we analyzed three cases of PBLH diurnal evolution under different meteorological and aerosol load conditions. In the presence of multiple aerosol layers, the employed algorithm exhibited high efficiency (r=0.9) in the attribution of PBLH, whereas weak aerosol gradients induced high variability in PBLH. A sensitivity analysis corroborated the stability of the utilized methodology. The comparison with CALIPSO observations yielded satisfying results (r=0.8), with CALIPSO slightly overestimating PBLH. Due to the relatively warmer and drier winter and, correspondingly, colder and rainier pre-monsoon season, the seasonal PBLH cycle during the measurement period was slightly weaker than the cycle expected from long-term climate records.

## 1 Introduction

The Planetary Boundary Layer (PBL) is the lowermost portion of the troposphere, which experiences a diurnal cycle of temperature, humidity, wind and pollution variations. PBL is a key component of the atmosphere and of the climate system, as it fundamentally affects cloud processes, as well as land and ocean surface fluxes. The PBL height (PBLH) is the most adequate parameter to represent the PBL. Therefore, it is usually required in numerous applications. For instance, in pollution-dispersion modelling, where the upper boundary of the turbulent layer acts as an impenetrable lid for the pollutants emitted at the surface. The PBLH also appears as a mixing scale height in turbulence closure schemes within climate and weather prediction models (Zilitinkevich and Baklanov, 2001). As air pollution becomes more severe due to economic development, particularly in developing countries (Wang et al., 2009), high temporal and vertical resolution observations of the PBLH are essential for weather and air-quality prediction and research. Moreover, the PBLH is related to the warming rate caused by enhanced greenhouse gases emissions (Pielke et al., 2007).

Several methods have been proposed to estimate the PBLH, utilizing vertically resolved thermodynamic variables, turbulence-related parameters and concentrations of tracers (Seibert et al., 2000; Emeis et al., 2004). Different methods for the determination of the PBLH from radiosondes have

been compared and the associated uncertainties have been estimated (Seidel et al., 2010; Wang and Wang, 2014). Restrictions of radiosondes refer to the coarse vertical resolution of standard meteorological data with respect to boundary layer studies as well as the smoothing due to the sensor lag constant bounded by the high ascent rate of the radiosonde (Seibert et al., 2000). Remote sensing systems such as aerosol lidar, microwave radiometer (Cimini et al., 2013), wind-profiling radar (Cohn

and Angevine, 2000) and Doppler wind lidar (de Arruda Moreira et al., 2018) are suitable for long-term measurements of various atmospheric quantities with high temporal resolution and can be used either independently or synergistically to retrieve the PBLH. Space-borne lidar systems provide the advantage of spatial coverage, although for studies focusing on a particular area of interest, measurements are constrained by the overpass frequency. Ceilometers are simple backscatter lidars, which entail less

operational cost. However, exploitation of their full potential is on an early stage with limited ceilometer-related studies (Münkel et al., 2007, Binietoglou et al., 2011, Wiegner et al., 2014). Ceilometers have high potential of contributing to PBLH climatology, within certain limits, but detailed investigation of open issues is still needed, as for example, the treatment of incomplete overlap. Additionally, no adjustments can be typically made by the user, contrary to the modified

Wavelet Covariance Transform (WCT) algorithm. Hence, improvements on layer detection algorithms are urgently needed to fully exploit the potential of ceilometers. In elastic and Raman lidar systems, the atmospheric aerosols are used as tracers and the PBLH is indicated by a gradient in the range-corrected lidar signal (Menut et al., 1999; Brooks 2003; Amiridis et al., 2007; Morille et al., 2007; Baars et al., 2008; Engelmann et al., 2008; Groß et al., 2011; Tsaknakis et al., 2011; Haeffelin et al.,

2012; Scarino et al., 2013; Summa et al., 2013; Korhonen et al., 2014; Lange et al., 2014; Bravo-Aranda et al., 2016). Weather and climate prediction models could alternatively be used to determine the PBLH, especially for strong horizontal inhomogeneity. However, inconsistencies in the definition of PBLH among the existing meteorological models also result in significant differences in its calculation (Tombrou et al., 2007).

New Delhi is one of the most densely populated cities and the fifth most populous city in the world according to United Nations population estimates and projections of major Urban Agglomerations (https://esa.un.org/unpd/wup/). It is surrounded by the Thar Desert to the west and the western Indo-Gangetic Plain to the north. Particulate air pollution in this area is assumed to originate from fossil fuel and biomass burning besides natural sources such as desert dust (Hedge et al., 2007; Ramanathan et al.,

2007a). The identification of the layer height within which pollutants are trapped is particularly important in this polluted area, since the largest and most persistent pollution haze covers an area of about 10 million $km^2$ over Southern Asia (Nakajima et al., 2007; Ramanathan et al., 2007a). Thus, vertically resolved observations are indispensable to reveal information regarding local air quality, climate change and human health related issues.

Despite the importance of the area under investigation, only few ground-based measurements of aerosol vertical profiles have been carried out, with most of the available data accessed during short

field campaigns (Lelieveld et al., 2001; Nakajima et al., 2007; Ramanathan et al., 2007a). In this study, we investigate PBLH characteristics over New Delhi, India, based on one-year long ground-based lidar measurements. The measurements were carried out from March 2008 to March 2009 in the framework of EUCAARI (European Integrated project on Aerosol Cloud Climate and Air Quality Interactions) project (Kulmala et al., 2011). The aim of this study is twofold; (1) to assess the efficiency and stability of the modified WCT technique in retrieving PBLH and (2) to compare the PBLH derived from ground-based lidar to independent data sources.

**2 Measurement site**

The lidar measurement site was located at Gual Pahari (28.43ºN, 77.15ºE, 243 m a.s.l.), which is situated in the Gurgaon district of Haryana state, about 20 km south of New Delhi, India (Hyvärinen et al., 2010; Komppula et al., 2012). The surroundings of the station represent a semi-urban environment with agricultural test fields and light vegetation. There were no major pollution sources, except for the road between Gurgaon and Faridabad about 0.5 km to the south-west of the station, while only electric-powered vehicles were allowed at the station area. Anthropogenic sources in the greater region comprised traffic, city emissions and power production (Reddy and Venkataraman, 2002a, b). Meteorological parameters were measured at the meteorological station of Safdarjung airport (28.58ºN, 77.21ºE, 211 a.s.l.), New Delhi, which is located 18 km NE of Gual Pahari and was the closest climatological site to the lidar measurement site.

During the measurement period, sunrise time varied between 5:45 and 7:15 LST, whilst sunset appeared between 18:15 and 19:15 LST. Solar noon appeared between 12:00 and 12:30 LST. Local time at New Delhi corresponds to UTC+5.5 h. From now on in this paper, UTC will be adopted, to facilitate the comparison between the lidar measurements and numerical simulations.

Temperature and precipitation patterns can potentially reflect the state of sensible and latent heat fluxes within the PBL as well as the exchange of moisture and momentum with the Earth's surface. Thus, climatologies of meteorological parameters can be considered a valuable tool for assessing the representativeness of PBLH seasonal cycle with respect to long-term measurements. Such a comparison is performed in Section 4.4 based on the 30-year anomalies of maximum temperature and accumulated precipitation (Figure 1).

**3 Methodology and Instrumentation**
**3.1 Ground-based lidar measurements**
**3.1.1 FMI-Polly[XT] lidar system**

The measurements were conducted with a six-channel Raman lidar called FMI-Polly[XT] (Finnish Meteorological Institute - Portable Lidar sYstem eXTedend). The lidar system was entirely remotely controlled via an internet connection, with all the measurements, data transfer and built-in device regulation being performed automatically. The instrument was equipped with an uninterruptible power supply (UPS) and an air conditioning system (A/C) to allow for safe and smooth continuous measurements. A rain sensor was also connected to the roof cover in order to assure a proper shutdown of the instrument during rain.

FMI-Polly[XT] used a Continuum Inline III type laser. The pulse rate of the laser was 20 Hz and it delivered energies of 180, 110 and 60 mJ simultaneously (with external second and third harmonic generators) at three different wavelengths, i.e. 1064, 532, 355 nm, respectively. A beam expander was used so as to enlarge the beam from approximately 6 mm to 45 mm. The remaining beam divergence after expansion was less than 0.2 mrad. The backscattered light was collected by a Newtonian telescope, which had a main mirror with a diameter of 30 cm and a field of view of 1 mrad. The output of the instrument included vertical profiles of the particle backscatter coefficient at three wavelengths, i.e. 355, 532 and 1064 nm (retrieved with the Klett method; Klett (1981) and Klett (1985)), extinction coefficient at 355 and 532 nm (retrieved with the Raman method (Ansmann et al., 1990; Ansmann et al., 1992) by using the Raman shifted lines of $N_2$ at 387 and 607 nm) and linear particle depolarization ratio at 355 nm. The system vertical resolution was 30 m and the vertical range covered the whole troposphere under cloudless conditions. This is sufficient for PBL studies considering the heights needed in this work. Engelmann et al. (2016) reports a maximum vertical range of 40 km, which depends on the capabilities (height bins) of the data acquisition. The FMI-Polly[XT] lidar system is described in more detail in Althausen et al. (2009) and Engelmann et al. (2016).

The incomplete overlap between the laser beam and the receiver field of view L-R (Laser-Receiver), restricted the observational detection range to heights above 200-300 m. This was partly counterbalanced by the overlap correction function. In this study, overlap corrections were performed at 532 nm following the methodology proposed by Wandinger and Ansmann (2002). During the measurement campaign, the L-R overlap was completed at 550-850 m, with the estimation of the full overlap height performed five times, since changes in the system could have affected the alignment between the laser beam and the receiving telescope optical axes.

During night-time, the configuration of FMI-Polly[XT] allowed the determination of the Residual Layer height (RLH). The study of Wang et al. (2016) which was performed at a station of similar latitude, Wuhan, China, revealed that the RLH lies mostly in the range 0.5–1.3 km, following a seasonal variation. Hence, for most of our night-time cases we considered that the lidar system detected the top of the residual layer, which contained the aerosol of the previously mixed layer. In particular, if a layer top more than 500m was detected between sunset and sunrise, it was associated with the RLH.

### 3.1.2 PBLH detection technique

The PBLH was derived from the 15-min averaged lidar backscatter signals at 1064 nm using the WCT method (Brooks, 2003) with modifications introduced by Baars et al. (2008). The algorithm of the WCT method was applied to 6-hour datasets. An overview of the lidar range-corrected signal was made available by TROPOS (Leibniz Institute for Tropospheric Research) and can be accessed at http://polly.rsd.tropos.de/?p=lidarzeit&Ort=21. The WCT method makes use of the assumption that the PBL contains much more aerosol load compared to the free troposphere and, thus, a strong backscatter signal decrease can be considered as the PBLH. The covariance transform $W_f(a,b)$ is based on the convolution of the range-corrected lidar signal and the related Haar function (Baars et al., 2008). This method was chosen because it allows larger adjustability than other techniques, as shown from

previous studies (Baars et al., 2008; Korhnonen et al., 2013). For instance, the gradient technique involves an ambiguity in the choice of the relevant minimum in the gradient that corresponds to the PBLH (Lammert and Bösenberg, 2005). A first modification by Baars et al. (2008) regarded the WCT threshold, which allowed the identification of significant gradients and the corresponding omission of weak gradients. The first height above ground at which a local maximum of $W_f(a,b)$ occurred, exceeding the selected signal decrease threshold, was defined as the PBLH. A second modification introduced by Baars et al. (2008) was related to strong gradients in the lower parts of the PBL (30-870 m) and the ability to exclude these parts from the lidar data evaluation. In this work, the applicability of the WCT technique under different meteorological and aerosol load conditions is discussed (Section 4.1) in the context of three case studies and the stability of the WCT algorithm is assessed as well (Section 4.2). Additional cases, where the importance of a proper threshold and cutting-off zone are discussed, can be found in Nakoudi et al. (2018). The WCT method also allows for the detection of clouds by means of a negative threshold. Baars et al. (2008) found that the cloud screening works well for a threshold of -0.1. The cloud base is given one height bin below the altitude at which $W_f(a,b)$ is lower than the chosen threshold value. The WCT method has also been applied for the detection of cirrus cloud base height over different geographical regions (Dionisi et al., 2013, Voudouri et al., 2018). Uncertainties in the retrieval of PBLH mainly originate from the lidar signal noise, which is less during night-time, the systematic error related to the estimation of the atmospheric molecular number density from the pressure and temperature profiles as well as the systematic error for overlap function. Furthermore, errors are introduced by the operation procedure such as signal smoothing and averaging by accumulating lidar returns. Detailed discussions on the overall relative errors of the Polly and Polly[XT] lidar derived aerosol properties can be found in Baars et al. (2016) and Engelmann et al. (2016).

Daily mean and maximum PBLH corresponds to convective hours (3:00-12:00 UTC). The hourly PBLH was calculated from the 15-min lidar observations by averaging of the three closest data points of the time considered (e.g. 12:00 hourly height would be the average of the three data points between 11:45 and 12:15). The seasonal cycle study was based on the classification proposed by the Indian Meteorological Department, i.e. winter (December-March), pre-monsoon or summer (April-June), monsoon (July-September) and post-monsoon (October-November) (Perrino et al., 2011). However, the PBLH seasonal cycle was examined during the winter, pre-monsoon and monsoon periods, as no sufficient data coverage was found during the post-monsoon period (Section 3.1.3). The PBLH growth period was determined following the guidelines of Baars et al. (2008). More specifically, the PBLH growth period began when the PBLH started to increase (typically 2-4 h after sunrise) and was completed when 90% of the daily maximum PBLH was reached (typically between 08:00 and 10:30 UTC). Concerning the daily evolution rate, this was determined through the slope of a linear fit to the hourly PBLH (between the start and the completion of the growth period). The evolution rate calculation was restricted to cases where at least 4 consecutive or 3 non- consecutive hourly values were available. Due to these restrictions, the evolution rate was determined for 44 days.

### 3.1.3 Data coverage

During the one-year long measurement campaign, FMI-Polly$^{XT}$ was measuring on 139 days. Due to technical problems with the laser, the data coverage from September to January was sparse. Furthermore, precipitation prohibited lidar measurements, since the lidar system had to shut down. Hence, sufficient data availability was achieved during 72 days. Multiple aerosol layers appeared mainly between March and May, whereas low clouds were present mostly in the monsoon period and both complicated PBLH detection. Additionally, some technical issues arose due to photomultiplier supersaturation and signal problems. A lack of a significant decrease in the backscatter profile was observed in only a few cases. The latter was a first indication that the modified WCT method can detect the PBLH efficiently, as long as the signal decrease threshold was tuned properly. The data coverage is presented on a monthly basis in Figure 2. The highest PBLH detection frequency was achieved in February, which can be attributed to favorable meteorological conditions, since low clouds appeared sparsely without any rainfall events.

**3.2 Space-borne lidar observations**

Cloud-Aerosol Lidar and Infrared Pathfinder Satellite Observation (CALIPSO) is an Earth Science observation mission that was launched on 28 April 2006. The vertical resolution of the CALIOP (Cloud-Aerosol Lidar with Orthogonal Polarization) system is 30 m. CALIPSO Level 2 aerosol layer product, provides a description of the aerosol layers, including their top and bottom height, identified by automated algorithms applied in the Level 1 data. Detailed description of the aforementioned algorithms can be found in Vaughan et al. (2004) and Winker (2006). In this study, CALIOP Version V4-10 dataset was used. Currently, no operational CALIOP PBL product is available.

More specifically we applied the CALIOP Level 2 Aerosol Layer Product, which provides information on the base and top heights of existing aerosol layers, reported at a uniform 5 km horizontal resolution. Leventidou et al. (2013) evaluated the daytime PBLH derived by Level 2 Aerosol Layer products over Thessaloniki, Greece, for a 5-year period, making the assumption that the lowest aerosol layer top can be considered as the PBLH. The aforementioned method was also applied over South Africa, revealing high agreement with ground based observations (Kohronen et al., 2014). During the measurement campaign, PBLH was accessed by the space-borne lidar CALIOP, within overpass distances of 20 and 101 km from Gual Pahari.

**3.3 WRF Atmospheric Model**

The Weather Research and Forecasting (WRF) model, Version 3.9.1 (Skamarock et al., 2008) was also applied in order to determine the PBLH. The simulation domain was centered at the lidar station in Gual Pahari and three domains with a respective horizontal resolution of 18 km, 6 km and 2 km were used, where the two inner domains are two-way nested to their parent domain. The third inner-most domain covers an area between 75.84-78.46º E and 27.38-29.52º N. The output is provided every hour. On the vertical axis, 37 full sigma levels resolve the atmosphere up to 50 hPa ($\approx$ 20 km AGL), with a finer grid spacing near the surface. In this study, the Yonsei University scheme (YSU) (Hong et al., 2006) in conjunction with the land surface model Noah (Chen and Dundhia, 2001) was used for the

estimation of PBL height. In addition, the Rapid Radiative Transfer Model (RRTM) scheme (Mlawer et al., 1997) for longwave radiation and the scheme of Dundhia (1989) for shortwave radiation were applied. A surface-layer scheme based on the revised MM5 similarity theory (Jimenez et al., 2012) as well as the Kain and Fritsch (1990, 1993) scheme for cumulus parameterization were used. For microphysics, the scheme proposed by Thompson et al. (2008) was considered. Regarding land use and soil types, the predefined datasets of Moderate Resolution Imaging Spectroradiometer (MODIS) with 21 land use classes were used. The initial and lateral boundary conditions were derived from the National Center for Environmental Prediction (NCEP) operational Global Fine Analysis (GFS) with 1 º x 1 º spatial resolution and were updated every 6 h. The Sea Surface Temperature (SST) was obtained from High Resolution Real-Time Global SST (RTG SST HR), with spatial resolution 0.083 º x 0.083 º which was renewed every 24 h.

In the YSU scheme, the top of the PBL under unstable conditions is determined as the first neutral level based on the Bulk Richardson Number (Ri) calculated between the lowest model level and the levels above (Hong et al., 2006; Shin and Hong, 2011). Under stable conditions, the Ri is set as a constant value of 0.25 over land, enhancing mixing in the stable boundary layer (Hong and Kim 2008), whereas it is a function of the surface Rossby number over the oceans, following the study of Vickers and Mahrt (2003). More specifically, the revised Stable Boundary Layer (SBL) scheme (Hong 2010) computes the exchange coefficients with a parabolic function with height, as in the mixed layer, in which the top of the SBL is determined by the Ri (Vickers and Mahrt 2004). This leads to a gradual-and not abrupt-collapse of the mixed layer after the sunset, due to the residual superadiabatic layer near the surface even in the presence of negative surface buoyancy flux. Within the frame of three case studies, the default simulated PBLH from WRF was used to justify the lidar PBLH.

## 4 Results and Discussion

### 4.1 Applicability of the WCT method: Case studies

It was found that in some cases the presence of multiple aerosol layers and low clouds can pose difficulties in PBLH detection (Section 3.1.3). However, these difficulties can be dealt with the use of proper WCT threshold and cut off values (Section 3.1.2). Three case studies of PBLH daily evolution were analyzed and evaluation with ancillary data sources was performed so as to investigate capabilities and limitations. First, the evolution of PBLH under cloudless conditions is discussed for 12 February 2009. Subsequently, a two-day case with a multiple aerosol layer structure is presented for 1-2 March 2009. Finally, the diurnal development of PBLH is investigated in the presence of low clouds for 29 June 2008.

### 4.1.1 Cloud free case: 12 February 2009

The PBL during 12 February 2009 was characterized by an almost constant daily growth rate (133 m/h between 06:00 UTC and10:00 UTC) with a maximum height of 950 m (Figure 3). No aerosol layers were observed in the free troposphere. Although gradients (yellow and red color) of aerosol content, appeared inside the PBL (06:00-12:00 UTC), the default signal decrease threshold (0.05) was efficient. However, later (12:00-18:00 UTC) in order to avoid strong gradients in the lower parts of the PBL,

higher threshold (0.08) was used in conjunction with cut-off heights (90 m). Furthermore, low aerosol load conditions were responsible for high variability in the derived PBLH (12:00-14:00 UTC).

During convective hours (05:00-12:00 UTC), WRF overestimated the PBLH mainly due to the simulated  neutral profile-virtual potential temperature at the surface similar to that around 1100 m AGL (differences < 0.5 K, not presented), resulting in an increase in the PBLH (Kim et al., 2013). It is worth mentioning that during the convective period FMI-Polly$^{XT}$ identified a light aerosol load activity at the altitude where the numerical models estimated the PBLH, with the WCT technique not detecting this activity due to the weakness of the aerosol gradients. During night-time model estimations yielded lower PBLH compared to lidar data.. During night-time model estimations yielded lower PBLH compared to lidar data. The low wind field produced by the WRF close to the surface (wind speed values up to 3 m/s in the first kilometer) and, thus, the lack of sufficient mechanical turbulence, can be related to the shallow nocturnal PBL. It should be noted that the measured PBLH is expected to depict, apart from any mechanically-driven layer during the stable and transition periods, the top of the previous day's residual aerosol layer, while the simulated PBLH from WRF refers to the height of the shallow mixed layer. Therefore, their difference is expected since they depict different layers. The overall correlation was satisfying (r=0.8).

**4.1.2 Case with multiple aerosol layers: 1-2 March 2009**

During the two-day period of 1-2 March 2009, a complex aerosol layer structure appeared in the free troposphere up to 3 km (Figure 4). However, appropriate modification of the signal decrease threshold and use of appropriate cut-off heights allowed for the detection of PBLH.  In order to avoid gradients in the lower parts of the PBL, the signal threshold was adjusted (0.03-0.08) within a 6-16% signal decrease, in combination with a 30-60 m cut-off zone.

On 1 March 2009, the transition period (02:00-05:00 UTC) was characterized by a slow PBLH development (14 m/h), whereas the PBLH evolution was more pronounced in the convective period (05:00-09:00 UTC) with a mean growth rate of 101 m/h. The maximum height (950 m) appeared at 08:45 UTC. On the next day, a stronger but slightly shorter PBLH cycle was observed, with a mean evolution rate of 187 m/h, reaching a maximum height (1010 m) at 08:15 UTC. This slight modification in PBLH development can be attributed to the combination of higher temperature and lower wind speed conditions during the second day. On the first day, WRF slightly overestimated PBLH during the transition period from CBL to RL (11:00-14:00 UTC), whereas on the second day an overestimation was observed during convective hours (9:00-12:00 UTC). During early morning and night, WRF underestimated PBLH, but the overall agreement with lidar observations was very satisfying (r=0.92 and r=0.95, on 1 and 2 March, respectively).

**4.1.3 Case with low clouds: 29 June 2008**

In this case broken cumulus clouds appeared between 600-1100 m (from 00:00 to 12:00 UTC). On average, a moderate PBLH evolution (86 m/h) was found, with a maximum height (1279 m) appearing at 9:15 UTC (Figure 5). Whenever clouds appeared below 1km, we made the assumption that the cloud base is an approach to the top of the PBL, however, it could be argued that the PBLH was at a higher

level, where diffuse aerosol layers were found. In addition, it was difficult to find an adequate signal decrease; the default threshold was used, while sensitivity tests with thresholds sensitive to weaker gradients yielded the same results. Hence, the algorithm exhibited decreased sensitivity, which can be attributed to the existence of diffuse aerosol layers. High PBLH was observed immediately after, due to a strong aerosol layer which sprawled to lower heights, either through dry removal or precipitation that evaporated before reaching the ground. Following a short rainfall period (13:30-14:30 UTC), the remaining aerosol kept being displaced downwards, creating strong gradients below 500 m. Moreover, the aerosol removal effect was clear (16:00-24:00 UTC) between 300 and 1000 m. Due to the low aerosol load, the detection of PBLH was complicated and, hence, accounted for the high variation in PBLH (16:00-24:00 UTC). WRF correlated well with FMI-Polly[XT] (r=0.74). During daytime, WRF slightly overestimated PBLH, while it should be noted that FMI-Polly[XT] identified intermittent aerosol gradients at the same altitude, which indicated turbulent activity.

### 4.2 Comparison with CALIOP L2 Aerosol Layer product

During the measurement period, 24 CALIPSO overpasses were available within 1° radius around Gual Pahari station. The Boundary Top Location algorithm SIBYL (Selective Iterated Boundary Locator) identified two to four layers (17 cases), while in the remaining cases no layers were identified. However, ground-based lidar observations were not available in all of the cases (only in 14). Furthermore, some cases (5) were excluded from the comparison as the detected layers were clearly above the typical PBL limits (higher than 3 km). The comparison of PBLH between ground-based and space-borne lidar (Figure 6) was fairly satisfying (r=0.84, statistical significant at 95% confidence level with 0.05 p-value), corroborating that the top of the first detected layer constitutes a good approximation of the PBL top in accordance to relevant studies (Leventidou et al., 2013; Kohronen et al., 2014). CALIPSO observations revealed slightly higher PBLH, since CALIPSO Layer Detection Algorithms in some cases possibly detected aerosol layers, which were transported aloft the PBL. In the majority of the analysed cases, the detected layers comprised dust layers with a few cases of dust-polluted dust and dust-polluted smoke mixtures, according to aerosol subtype classification. Based on the analyzed cases, it was found that the overpass distance (here 20 and 101 km) from the lidar station and time difference between the measurements did not affect the agreement of the PBLH. Furthermore, the layer top altitude did not appear to change systematically between daytime and night-time. However, the small number of measurements does not allow us to generalize these findings. Hence, longer measurement periods or more extended comparison to ground stations is needed in order to draw more robust conclusions.

### 4.3 Statistical Analysis
### 4.3.1 Diurnal Cycle of PBLH

Although night-time PBLH is not taken into account for the statistical analysis of seasonal PBLH (Section 4.4.2), nocturnal PBLH is considered here in order to investigate the diurnal evolution of PBLH. In winter, the PBLH cycle as defined by FMI-Polly[XT], reached its maximum (1028 ± 292 m) at

11:00 UTC, while the CBLH evolution was completed two hours earlier (Figure 7b). In pre-monsoon period, the PBLH growth as derived from lidar was completed three hours prior to PBLH maximization (1249 ± 536 m) (Figure 7c). In monsoon, FMI-Polly$^{XT}$ revealed a fairly smooth PBLH cycle (Figure 7d). The maximum PBLH (1192 ± 187 m) was observed earlier, compared to winter and pre-monsoon, with high PBLH persisting for a couple of hours afterwards. Turbulence produced by convection usually reaches maximum values immediately after the solar noon, but further growth of PBLH cannot be sustained for a long period. Nevertheless, PBL did not appear to collapse immediately afterwards, probably due to the remaining turbulent fluxes.

### 4.3.2 Daily mean and maximum PBLH

In this Section we statistically analyze the seasonal mean and maximum PBLH cycle as observed from lidar measurements in conjunction with the seasonal cycle of mean and maximum temperature. The seasonal mean PBLH was found at 695 ± 146 m during winter, 878 ± 297 m during the pre-monsoon period and 1025 ± 296 m during the monsoon. The seasonal average maximum PBLH was determined at 1191 ± 516 m during winter, 1326 ± 565 m during the pre-monsoon period and at 1361 ± 350 m during the monsoon. In general, the PBLH seasonal cycle followed the temperature cycle very well. The temperature cycle of the measurement days was fairly representative of the whole 2008-2009 seasonal cycle, with the temperature distribution being similar to the distributions of the whole seasonal periods. During the measuring period, a mean temperature of 21 ± 4 °C was found in winter, 27 ± 3 °C in the pre-monsoon and 30 ± 2 °C in the monsoon season while the seasonal average maximum temperature was recorded at 29 ± 5 °C, 33 ± 4 °C and 35 ± 2 °C accordingly. Nevertheless, it should be mentioned that the seasonal cycle of PBLH over Gual Pahari was weaker than the climatologically expected. More specifically, the smoother PBLH cycle could be explained in terms of maximum temperature and cumulative precipitation anomalies (Figure 1). During winter, average maximum temperature was 5 °C higher than the climatological one, while total precipitation was lower (10 mm). On the other hand, during pre-monsoon the average maximum temperature was lower by 5 °C than the corresponding climatological record, along with a significantly higher seasonal accumulated precipitation (205 mm). The latter is also related to the fact that in 2008 (16 June) one of the earliest monsoon onset dates (rainfall data since 1901) was recorded (Tyagi et al., 2009). Compared to another site with similar surroundings and solar cycle in Elandsfontein, South Africa, the annual average PBLH was lower in Gual Pahari (866 m; 1400 m in Elandsfontein) with less seasonal variability (Kohronen et al., 2014).

In winter, the daily mean PBLH distribution was narrower (in majority between 600 and 900 m) compared to the pre-monsoon and monsoon seasons (mostly between 900 and 1200 m). Following a similar pattern, the daily maximum PBLH was rather confined in winter (in majority between 900 and 1200 m) with a significantly broader spectrum (between 600 and 1800 m) in pre-monsoon and monsoon. The highest inter-seasonal variability was exhibited during pre-monsoon, which could be attributed to meteorological conditions. The pre-monsoon season comprised days with heavy rainfall and days with hardly any precipitation, which can potentially explain the broad distribution of daily

mean PBLH (251-1191 m). In winter large inter-seasonal variability of maximum PBLH was observed,
which can be possibly attributed to the broad inter-seasonal temperature range (20 °C - 36 °C).

### 4.3.3 Daily evolution rate of PBLH

During the measurement period, daily evolution rates were mostly within 100-200 m/h but lower rates
(29-100 m/h) were observed as well. In winter, daily growth rates presented a slightly broad
distribution (mostly between 100 and 200 m/h) with a mean evolution of $157 \pm 81$ m/h (Figure 8). In
pre-monsoon, slightly higher growth rates were observed (mainly within 100-300 m/h), with an
average of $206 \pm 134$ m/h. Additionally, rates between 0-100 m/h and 500-600 m/h were observed,
following the pattern of high inter-seasonal variability which was revealed during the pre-monsoon
season (Section 4.4.2). In the monsoon, evolution speeds were slightly lower ($121 \pm 67$ m/h) compared
to the pre-monsoon season. The distributions of daily growth rate during pre-monsoon and monsoon
show similarities. In order to examine whether the distributions are statistically different we applied the
two-sided Wilcoxon rank sum test (Wilcoxon, 1945; Wilcoxon and Wilcox 1964). The test yielded that
the two distributions are statistically different at the 95% significance level. Hence, the differences in
the growth rates between pre-monsoon and monsoon could be possibly related to the weaker diurnal
PBLH cycle that was found during the monsoon (Figure 7c). In addition, the different precipitation
patterns, with less precipitation during pre-monsoon, could attribute to the different growth rates. In
Elandsfontein, maximum rates (between 120-320 m/h) were reached during spring, September-
October, (Kohronen et al., 2014) a period that exhibits strong similarities with the pre-monsoon season
in India.

### 5 Summary and Conclusions

In this study, one year long ground-based lidar measurements were used to retrieve PBLH over Gual
Pahari, New Delhi. The feasibility of deriving PBLH with the modified WCT technique was
investigated and the respective results were compared to independent sources.

In support of previous work (Baars et al., 2008; Korhonen et al., 2014), it was found that the modified
WCT method exhibited satisfying efficiency under different meteorological and aerosol load regimes.
On a case with elevated aerosol layers, significantly good performance was revealed, even when the
layers were injected into the PBL. Such layers have been reported in literature as a major challenge in
the attribution of the PBLH especially during night-time (Haeffelin et al., 2012). PBLH determination
was complicated in the presence of diffuse aerosol layers. Low aerosol load, observed mainly during
morning or afternoon transitions, also represents a condition for uncertain determination of PBLH
(Haeffelin et al., 2012). Sensitivity analysis revealed stable performance of the WCT algorithm, with
the exception of elevated layers and PBL internal gradients, which affected the results when specific
thresholds were applied. Higher thresholds appeared to be more sensitive towards detecting lofted
layers.

 In the context of the aforementioned cases, WRF model overestimated PBLH in the daytime, while an
underestimation was observed in the night-time. The understanding of turbulence in nocturnal SBL and
its parameterization is rather slow and not well established in NWP models (Mahrt et al., 1999; Beare

et al., 2006; Hong 2010). In this study, this is partly addressed by the revised SBL scheme that retains the turbulent levels so as to avoid the abrupt collapse of the mixed layer after the sunset by using the exchange coefficients. However, the fact that neither anthropogenic heat sources nor heat storage in buildings were included in the simulations could also explain the model underestimation. Furthermore, it should be noted that the measurements often depict different layers from the simulated ones, as in the case of the residual aerosol layer. Detailed studies of the nocturnal boundary layer, which require changes in the lidar configuration, such as employment of a near range and a far range telescope (Engelmann et al., 2016) can improved the overall consistency in PBLH retrieval approaches between the model and lidar observations.  Satellite lidar observations correlated well with ground-based measurements, yielding higher PBLH due to the detection of lofted aerosol layers in some of the cases. These layers can potentially blanket the PBL and, hence, may strongly attenuate the emitted laser beam. More comparisons with ground-based lidar observations are needed to support that the top of the 1st layer is indicative of the PBLH.

During the rainy season of monsoon, the diurnal cycle of PBLH was weaker and its evolution was completed earlier. A relatively warmer and drier winter and, respectively, a colder and rainier pre-monsoon were observed compared to climatological records. These meteorological patterns could account for the observed PBLH cycle, which was rather indistinct compared to the cycle expected from long-term climate statistics. Daily evolution rates of 29-200 m/h were mainly observed, with lower rates during the rainy season of monsoon.

Future studies are necessary in order to better understand the factors that modulate the exchange of moisture, heat and momentum between the surface and PBL and, consequently, affect the comparison of modelled PBLH with observational data. In addition, the relative contribution of the various PBL dynamics drivers, under different aerosol load and meteorological regimes, needs to be further investigated. The feasibility of applying the modified WCT method in simpler lidar systems such as ceilometer and Doppler lidar, should be assessed. These systems entail less operational cost and, thus, exhibit good potential for determining the PBLH and evaluating weather prediction and pollution dispersion models on an operational basis. In recent years, significant effort has been made towards the establishment of ceilometer networks by national weather services and other agencies over Europe with the aim to build up a framework for real-time applications and improvements of air quality and weather prediction by assimilation of ceilometer data (Haeffelin et al., 2012; Wiegner et al., 2014). Analogous efforts are currently in progress over different parts of India, like in the states of Maharashtra and Kerala and in the union territory of Delhi (Sharma et al., 2016; Babu et al., 2017; https://www.lufft.com/projects/several-lufft-chm-15k-ceilometer-projects-in-india-529/).

**Appendix A: Sensitivity Analysis of the WCT threshold**

In cases of elevated layers or aerosol gradients within the PBL, it has been revealed that the signal decrease threshold needs to be properly adjusted (Section 4.1). In this study, we adapted the threshold (t) so that the WCT algorithm was allowed to identify signal gradients in the order of 6-16% (t=0.03-0.08, correspondingly). In this Section, we investigate the effect of the WCT threshold on the estimated

PBLH. For this reason, we performed a sensitivity analysis modifying the signal decrease threshold for the case of 2 March 2009, where elevated layers were injected into the PBL.

The overall performance of the WCT technique was stable (Figure 9), with the threshold affecting the results in only a few cases. When the lowest and more sensitive to detect weak layers threshold (0.03) was applied, a thin aerosol layer (around 1300 m) was identified (see Figure 4). At this time (07:00 UTC), increased thresholds (0.04-0.08) detected a stronger elevated layer (approximately at 2 km). The lowest threshold was also more efficient when gradients appeared inside the PBL (around 17:00 UTC), with the higher thresholds yielding increased PBLH by approximately 300 m. When the elevated layers were characterized by higher aerosol load (18:00-19:00 UTC), lower thresholds (0.03-0.05) performed better as well, with the higher ones identifying stronger layers (around 1 km). Thus, the PBLH deviation, introduced by the modification of the WCT threshold, appeared to depend on the altitude of internal gradients or elevated layers. However, in the early morning (00:00-03:00 UTC), where the convective activity was not initiated yet, a minor fluctuation (30 m) was observed, related to the algorithm's sensitivity towards aerosol content gradients.

An adequate threshold adaptation also affected the agreement with the modelled PBLH. More specifically, it is shown (Figure 9) that during cases where the applied threshold induced a deviation from the smooth PBLH evolution, the disagreement with modelled PBLH increased as well. Besides, the agreement with the simulated PBLH appears to depend on the altitude of the atmospheric features (internal or elevated aerosol gradients) that affect the performance of the WCT algorithm.

## Appendix B: Statistical Indicators

Pearson Correlation Coefficient

$$R = \frac{\sum_{i=1}^{N}(O_i - \overline{O}) \cdot (M_i - \overline{\overline{M}})}{\sqrt{\sum_{i=1}^{N}(O_i - \overline{O})^2} \cdot \sqrt{\sum_{i=1}^{N}(M_i - \overline{M})^2}}$$

$M_i$ denotes predicted values from models, while $O_i$ stands for observations at i, respectively. N is the number of samples.

*Acknowledgements*

This work (the campaigns) was partly funded by the European Integrated project on Aerosol Cloud Climate and Air Quality Interactions, EUCAARI. This work was supported by the Cy-Tera Project (NEA YPODOMH/STRATH/0308/31), which is co-funded by the European Regional Development Fund and the Republic of Cyprus through the Research Promotion Foundation.

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

Zilitinkevich, Sergej, Baklanov, Alexander: Calculation of the Height of Stable Boundary Layers in Operational Models. Danish Meteorological Institute, Copenhagen, 2001.

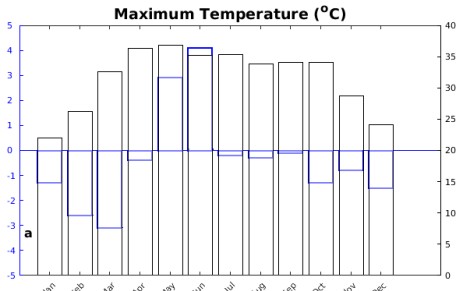 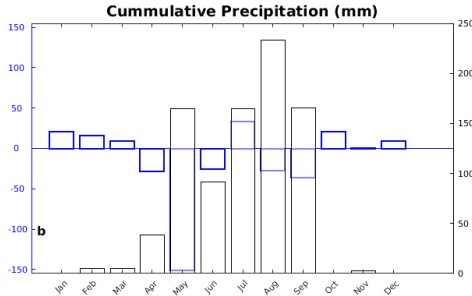

**Figure 1:** Maximum temperature and cumulative precipitation during the measurement campaign (black) and anomalies (blue) at New Delhi on a monthly basis. Anomalies represent difference between the climatological values and the corresponding values during the measurement campaign. Climatological values were obtained from World Meteorological Organization ( http://worldweather.wmo.int/en/city.html?cityId=224) for the site of Safdarjung airport.

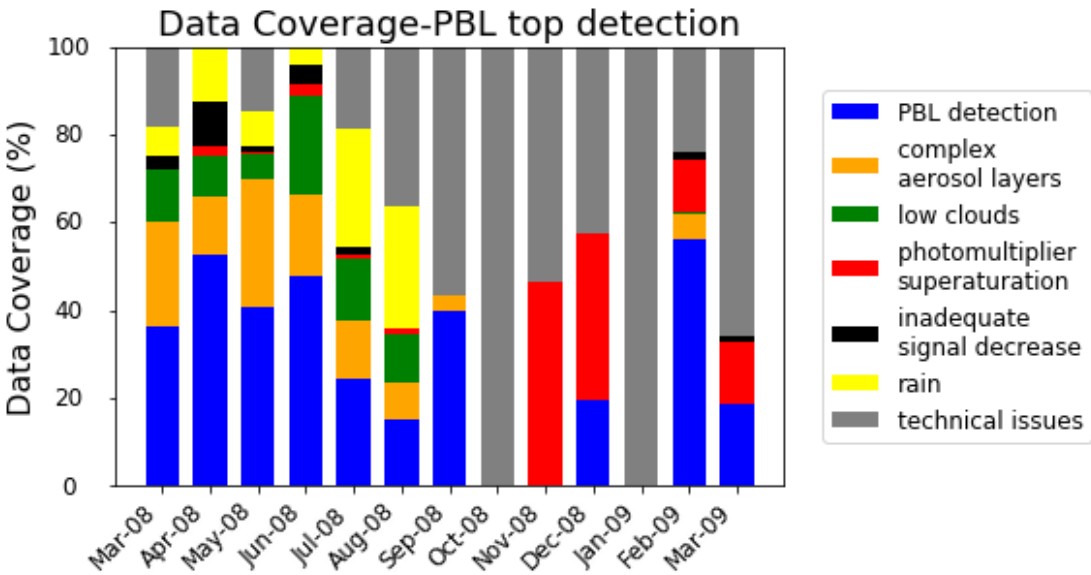


**Figure 2:** Data coverage of lidar measurements calculated with respect to total convective hours (from 4 h after sunrise to 1 h before sunset) during the measurement days of the campaign.





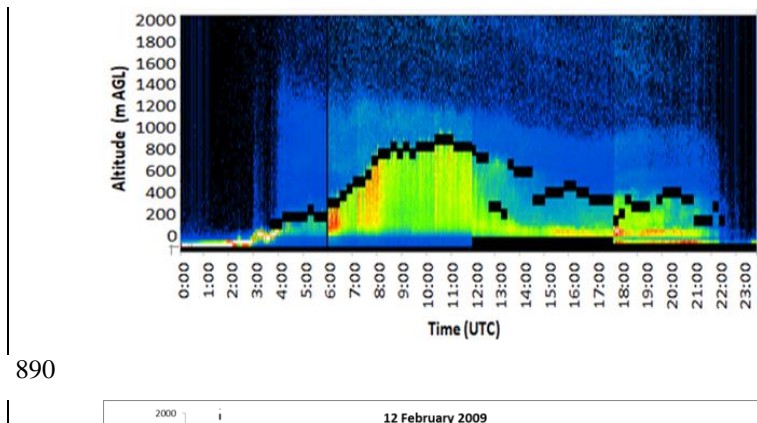

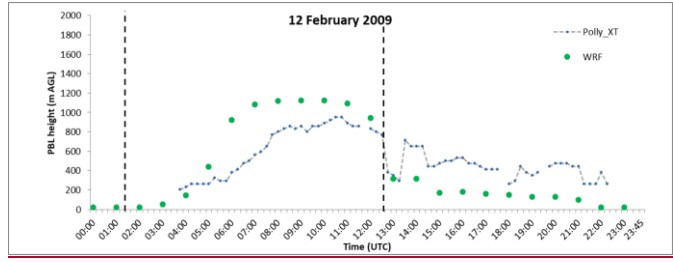

**Figure 3:** Evolution of PBLH observed on 12 February 2009. Range-corrected signal (top) at 1064 nm as measured with FMI-Polly[XT]. Black lines indicate 15-min PBLH, while black zones in the lower part of the figure indicate the extent of the signal cut-off area. The colorscale is normalized on a 6-hour basis, with red and yellow indicating higher aerosol load, while green and blue lower load, respectively. PBLH (bottom) as given by the FMI-Polly[XT] and WRF (vertical lines indicate sunrise and sunset times).

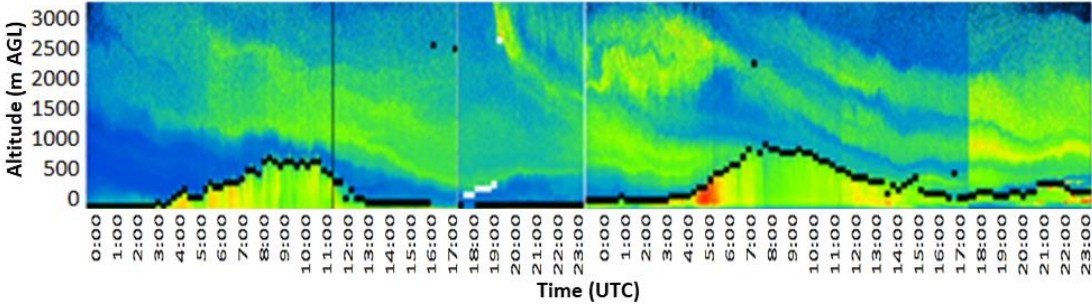

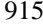

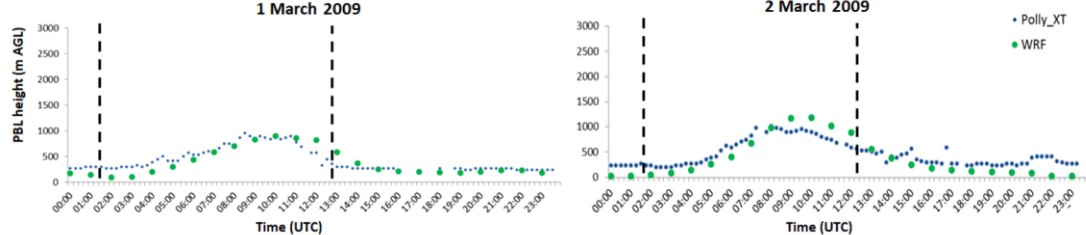

**Figure 4:** Same as Fig. 3 except for 1-2 March 2009. White horizontal lines (top) indicate 15-min cloud base height.

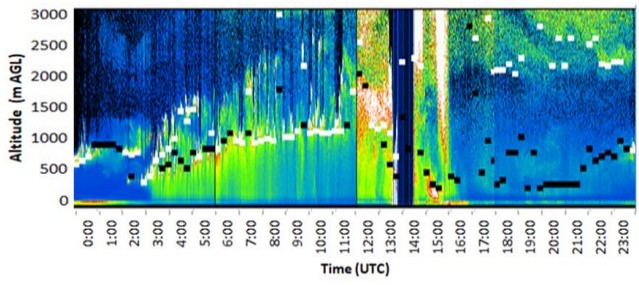

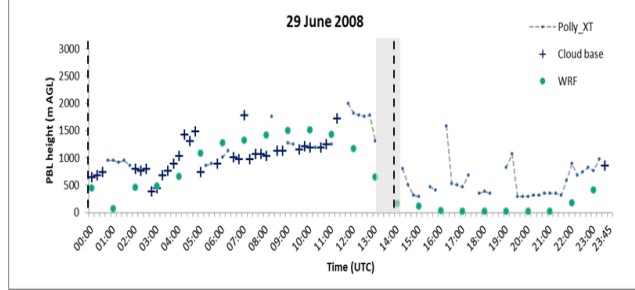

**Figure 5:** Same as Fig. 3 except for 29 June 2008. Grey shading (middle) indicates rainfall.







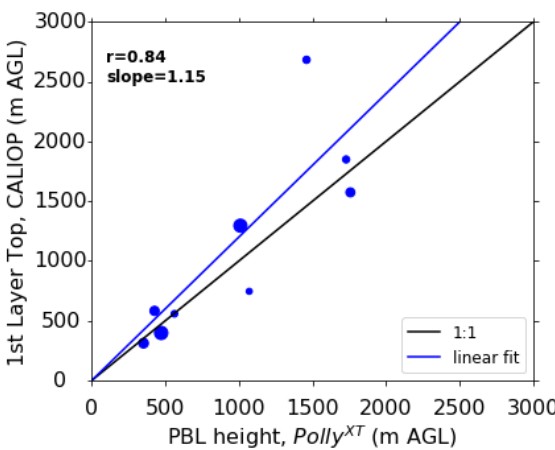


**Figure 6:** PBLH comparison for FMI-Polly[XT] and CALIOP. The heights given by CALIOP have been corrected with elevation. The markersize is proportional to the overpass distance from the ground-based lidar.




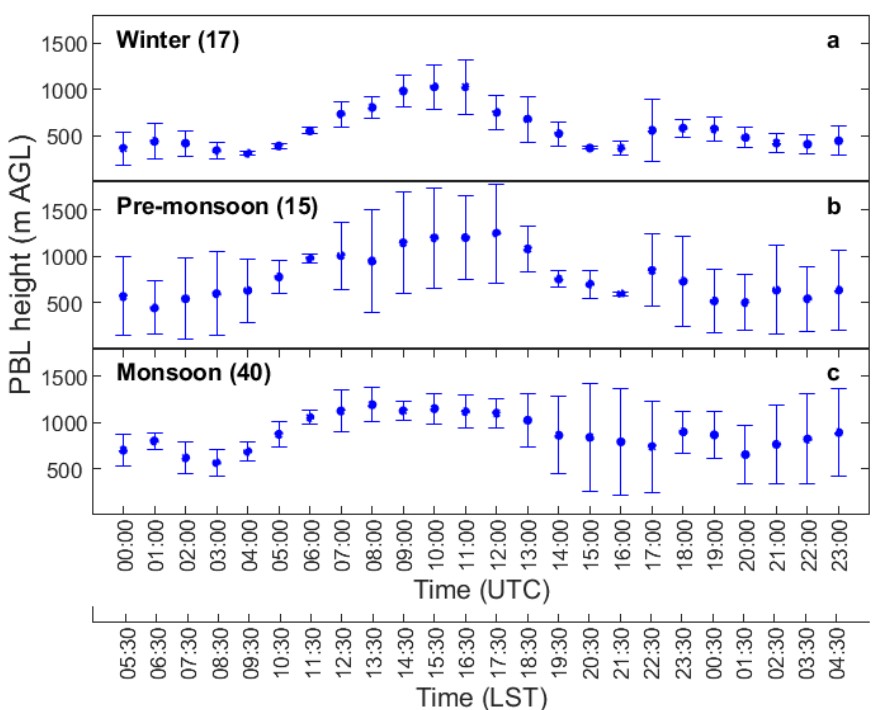

**Figure 7:** PBLH average diurnal cycle in Gual Pahari according to FMI-Polly$^{XT}$ during Winter (a), Pre-monsoon (b) and Monsoon season (c). Numbers indicate data availability.

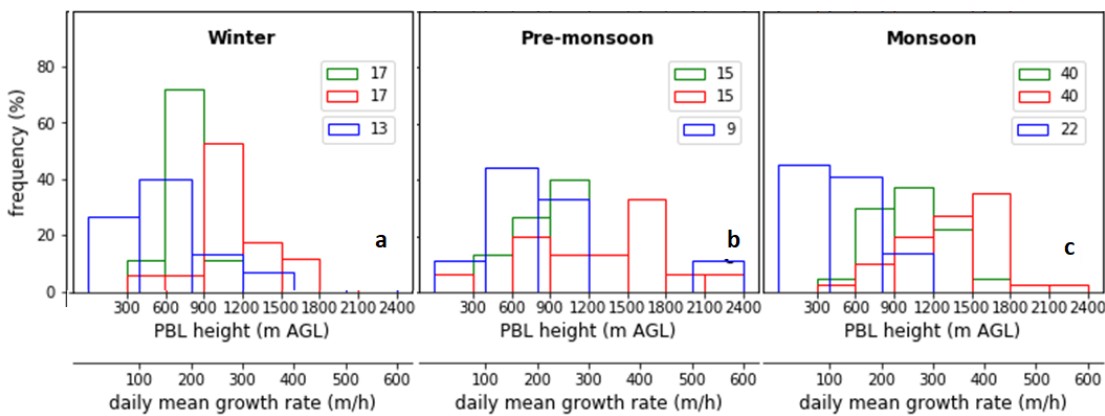

**Figure 8:** Frequency distribution of daily mean (green) PBLH, daily maximum PBLH (red) and daily mean growth rate (blue) as calculated during the winter period (a), the pre-monsoon season (b) and the monsoon period (c). Numbers indicate data availability.

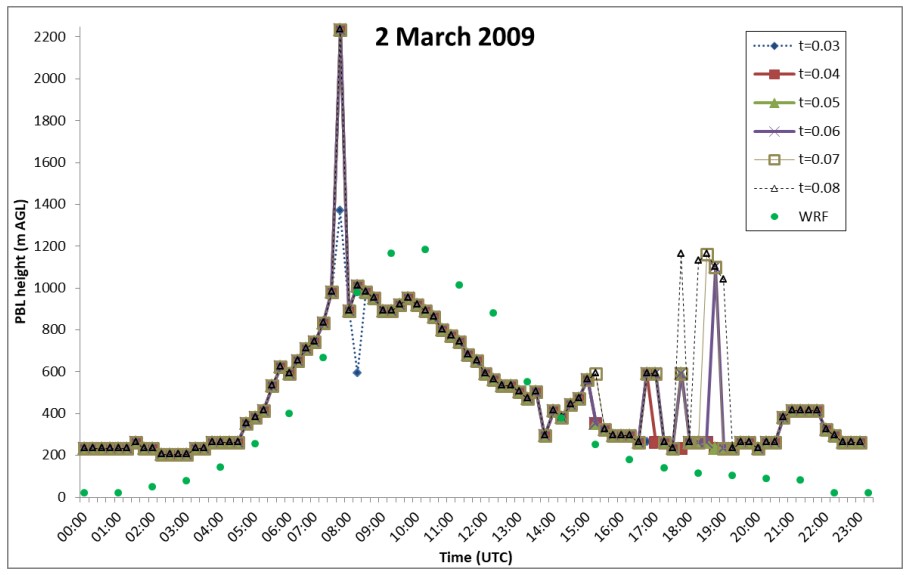

**Figure 9:** Sensitivity analysis of the WCT method for the case of 2 March 2009. PBLH was estimated by FMI-Polly[XT], after modification of the WCT threshold, and by WRF model.

1030

1035