# Peer review of "Planetary boundary layer height by means of lidar and numerical simulations over New Delhi, India"

_Atmospheric Measurement Techniques, 2018_

## Referee Comment (RC1) · Anonymous Referee #3 · 16 Jan 2019

**General comments:**

This paper presents a study on the planetary boundary layer height (PBLH) based on lidar measurements collected in New Delhi, a populous city facing severe air pollution. The detection of PBLH based on lidar data is performed using the Wavelet Covariance Transform (WCT) method, which is developed by Baars et al., 2008. The derived PBLH is then compared with the results obtained from other sources, i.e. radiosonde, CALIPSO and models (ECMWF and WRF). The authors did a lot of work in the data treatment, comparison and statistical study. The content of this paper generally falls into the scope of AMT journal, but major modifications are still needed.

This paper looks more like a report, but not like a scientific article. The authors present mostly what has been processed and some obvious results according to it. It is not

well structured and the content is not concise. The authors need to re-organize the materials and improve the manuscript to better support the topics they want to present in this paper.

**Particular comments:**
The application of WCT method is one the main topics of this paper. The first two case studies show satisfactory performance of the WCT method. However, there is not an overall comparison of PBLH between lidar PBLH detection and model estimation, although that exists for radiosonde (Figure 6) and CALIPSO product (Figure 7). The radiosonde measurement is very sparse and with low resolution, and the CALIPSO data are too few. So I consider that comparison of PBLH between lidar detection and model estimation should be of more importance, but it is not shown.

There is a lengthy comparison between lidar PBLH detection and other datasets. Maybe the authors are trying to 'grab' too many materials: radiosonde, CALIPSO product, ECMWF and WRF, as a result the 'story' becomes complicated. As the authors have mentioned in the manuscript, the definition of PBLH in different data sources is different by nature. Apart from the definition of PBLH, the comparison is also limited by other factors such as the temporal, spatial resolution of the data sources, data abundance and the spatial displacement. Comparing these results will inevitably bring some disagreements and also agreements. I think it is not necessary to show all of them, just select one or two datasets to compare and it will simplify the organization of this paper. For example, when comparing the T, $\theta$ and RH profile, radiosonde and WRF data are used; when comparing the diurnal evolution, lidar and ECMWF data are used, what is the purpose of involving the two models? Too many comparisons are made but some of them lack constructive conclusions.

Moreover, the manuscript needs to be strongly improved. The authors should avoid excessive details in the manuscript. Main messages, not all the details contained in the figures or tables should be addressed in the text. For example, numbers in 'data coverage' section do not carry so important meaning and do not need so many

words. The figures showing the T, $\theta$ and RH profiles, contain repeating information with Table 2 and Table 3, maybe only one such figure is enough...Please highlight the important and relevant information. The abstract and conclusion should be re-written after corrections of the content. The conclusion needs to be more conclusive and there is not need to go into details or repeat what has been said.

Additionally, I miss more scientific meaning of this work. The PBLH is certainly important for studying the atmospheric process in the boundary layer and for the air quality model or weather model. Such information will add value to this paper.

**Other comments:**

- P3, L85: Please check the latitude and longitude of the two stations.

- P3, L93-108: This paragraph presents very detailed information about the meteorological conditions of the observation site, but I think the messages that authors have presented are not well selected and structured. Details will bring details that need to be well commented (ex. differences between your measurements and WMO data), making the readers lost. The authors could shorten this paragraph and present the information that serves the topics and results that will be present in the following sections.

- P3-P4: I think the lidar system deserves more 'words', even it has been presented in other papers. Main messages should be addressed to the readers:

    - Elastic and Raman wavelengths
    - Telescope, divergence, overlap. . .
    - Data quality

- P4, L145: "A certain range of signal is cut to avoid strong gradients in the lower range", so how do you choose the range to be cut? Is it automatic?

- P4, L152: Comment the abbreviation 'L-R'; and the paper "Wandinger and Ansmann, 2002" is not included in the bibliography.

- P5: The Subsection '3.1.3 Data coverage' is very 'dry' and not so interesting. It does provide some information, but it is too detailed with unnecessary technical issues, and some numbers are not well defined so I got confused and sometimes had to redo these simple calculations. I think this paragraph could be shortened.

- P8, L203: And in Figure 3, Please comment the PBLH close to 0 m, why do the models produce such low values? There are two T profiles in the figure at right, is it a mistake? What are the white squares in Figure 3 upper panel?

- P9, L311: I miss a clear and quantitative difference between the PBL and RL. In Figure 3, there is a noticeable decrease of lidar signal at 1000-1200 m, do you consider it as the residual layer? Please also check the color scale of Figure 3. I saw a clear discontinuity at 06:00 UTC, 12:00 UTC and 17:30 UTC, are the lidar signals ploted with the same color scale? Moreover, the width of the black zone near the surface in Figure 3 is also changing, what does it mean? The authors should be more careful in preparing scientific figures.

- P9, L315: PBLH=435 m is not found in Figure 3, check the text, table 2 and Figure 3.

- P9, L333, define PBL cycle, when does this cycle start and end?

- P10, L354: which data did you use to derive this 553m/h? WRF?

- Figure 4, and Figure 5: Again, check the color scale please. The discontinuity is quite obvious.

- P10, Subsection 4.2.2: The data points are so few, and I do not think it is enough for a statistical study. "Based on the analyzed cases, it was found that the over-pass distance (here 20 and 101 km) from the lidar station and time difference between the measurements did not affect the PBL heights. " This conclusion does not convince me, because the dataset is so small and cannot represent the spatial and temporal variability.

- P11: PBLH diurnal Cycle might be more specific than PBL diurnal Cycle, because the authors investigated only the PBL height, not other parameters in the PBL.

- P12, L426: why is the comparison made only between lidar detection and ECMWF, how about WRF?

- P14, Section 5: not relevant and too short to be a section

- P15 conclusion: this section is long and not conclusive, and it is repeating what have been said previously.

---

## Referee Comment (RC2) · Anonymous Referee #2 · 16 Jan 2019

General comments

The paper presents the results of one year long ground-based lidar measurements used to analyze PBL height variability over Gual Pahari, New Delhi. PBL heights are obtained utilizing the modified Wavelet Covariance Transform (WCT) method. These results are firstly compared to four independent datasets: radiosonde data, CALIPSO satellite observations and ECMWF and WRF atmospheric models. Secondly, statistical and seasonal analyses are performed. The study highlights that for the detection of PBL height, the modified WCT method can be applied under different meteorological and aerosol load regimes. The comparison between the different datasets shows high discrepancies for nighttime PBL height estimation and a relatively good agreement for daytime estimation (e.g. lidar vs rds).

[Figure]

Whereas the study does not reveal significant new findings about the PBL height estimation, the main results are of interest. However, some sections of the paper lack of an accurate and more detailed description and needs to be completed. In particular, due to the small number of the considered samples, statistical significance should be added in the data analysis (e.g. lidar vs CALIOP comparison). Scientific significance of the study needs to be clarified in the paper. Comparison results should be discussed in more details taking into account the impact of different factors such as: the choice of proper WCT threshold and cut off values, differences in the ECMWF and WRF model vertical and horizontal resolution, lidar instrumental configuration (overlap and daytime vs nighttime configuration), error sources analysis, etc. . . Overall, to significantly strengthen the paper, these issues should be addressed. Thus, I recommend the publication of the manuscript after major revisions, according with the following observations.

Major comments:

Introduction

- Lines 45-47, I suggest adding more details about the detection of PBL height using radiosonde measurements and other instruments (e.g. microwave radiometers) with the appropriate references.

- Lines 47-54, not all the lidar systems can provide continuous, systematic and quantitative measurements of atmospheric aerosol profiles. Please clarify this aspect and expand this section, adding strengths and weaknesses between space-borne and ground-based systems and between lidar ceilometers and research lidars.

- Lines 73-76, please add more details on each section of the paper.

Section 2

- Lines 93-104, to understand if the anomalies in temperature and precipitation are significant, you should also report in the discussion and in figure 1, the standard deviation

of the measured and climatological values.

Section 3

3.1.1

- Lines 121-122, please specify the differences between nighttime and daytime configurations and the corresponding vertical sounding ranges.

- Lines 125-127, please add more details about the overlap factor of the system. Is a correction applied? Which is the height of full overlap? Since incomplete overlapping could hamper the PBL height detection, this characteristic should be well specified.

3.1.2

- Lines 143-145, and 151-154, see the comment for lines 125-127. The overlap characterization should be discussed in details in section 3.1.1.

- Lines 148-150, the WCT method was also applied for the detection of cirrus clouds height base (Dionisi et al., 2013, ACP) where a sensitivity study was made to fix a proper threshold. Please add this reference.

3.3

- Line 220, please specify the CALIOP version dataset used in this study.

- Lines 225-227, please specify if the used CALIOP overpasses are nighttime or daytime measurements.

Section 4

4.1

Please specify the impact of the WCT threshold on the results. A sensitivity study could be of help to interpret the results. Which is the associated error to the estimated PBL height? Is there any correlation between the magnitude of this error and the agreement between the different datasets? The effect of the different model horizontal and vertical

resolution should be also added in the discussion.

- Lines 313-317. Please explain the difference on PBL heights retrieved by radiosonde and WRF model.

4.2.1

- Lines 384-386. The results of the PBL height comparison between lidar and rds during daytime seems due to the sum of two opposing effects: the overestimation and underestimation of PBL height by rds, respectively. In fact, few points are along the 1:1 black line of figure 6, with two clouds of points on the right and on the left of the 1:1 line. This is confirmed by a significant but not very satisfying correlation ($R2 = 0.46$). Please explain this effect and rephrase this section. Please add the statistical significance of the comparison

- Lines 386-390. The impact of the different technical setups of the lidar system on the results should be quantified.

4.2.2

- Lines 405-409. Please add the statistical significance of the comparison.

- Lines 409-413. The number of considered cases is probably too small to generalize these results. Is there any noticeable difference between the different aerosol type of detected layers or between nighttime and daytime comparisons?

4.3

Is it possible to add in this analysis the mean diurnal PBL evolution estimated through WRF simulations?

4.3.1 Is there an explanation for the ECMWF overestimation (Polly underestimation) of PBL top height during convective hours for Winter and Pre-monsoon seasons and the ECMWF underestimation (Polly overestimation) for Monsoon season? The good agreement found at 12 UTC should also be highlighted.

4.3.3 The considered cases for this analysis are only 44 whereas for the previous section the number is higher (72). Please explain this difference. Statistical significance should also be specified. The measured differences in the growth rates between pre-monsoon and monsoon season can be attributed to a real signal or the poor significance of the sample does not allow any physical explanation? Please clarify these aspects.

Section 5

I'm not sure that this section is bringing any relevant information. Please motivate this comparison with further details and results or remove the section.

---

## Author Response (AR1)

Formatted

**Planetary boundary layer height by means of lidar and numerical simulations over New Delhi, India**

Konstantina Nakoudi[1,2], Elina Giannakaki[1,3], Aggeliki Dandou[1], Maria Tombrou[1], Mika Komppula[3]

[1]Department of Environmental Physics and Meteorology, Faculty of Physics, University of Athens, Greece
[2]Alfred Wegener Institute, Helmholtz Centre for Polar and Marine Research, Potsdam, Germany
[3]Finnish Meteorological Institute, Kuopio, Finland

*Correspondence to*: K. Nakoudi (knakoudi@phys.uoa.gr)

**Abstract.** In this work, the  height of the Planetary Boundary Layer (PBLH) is investigated over Gual Pahari, New Delhi, for almost a year. To this end, ground-based measurements from a multi-wavelength Raman lidar, were used. The modified Wavelet Covariance Transform (WCT) method was utilized for PBLH retrievals. Results. were compared to  data from, Cloud-Aerosol Lidar and Infrared Pathfinder Satellite Observation (CALIPSO)  and the Weather Research and Forecasting (WRF) model. In order to examine the difficulties of PBLH detection from lidar, we analyzed three cases of PBLH diurnal evolution under different meteorological and aerosol load conditions. In the presence of  multiple aerosol layers, the employed algorithm exhibited high efficiency (r=0.9) in the attribution of PBLH, whereas weak aerosol gradients induced high variability in PBLH. A sensitivity analysis corroborated the stability of the utilized methodology. The ~~PBL height determination. Good agreement with the European Center for Medium range Weather Forecasts (ECMWF) and the Weather Research and Forecasting (WRF) estimations was found (r=0.69 and r=0.74, respectively) for a cumulus convection case. In the aforementioned cases, temperature, relative humidity and potential temperature radiosonde profiles were well compared to the respective WRF profiles. The Bulk Richardson Number scheme, which was applied to radiosonde profile data, was in good agreement with lidar data, especially during daytime (r=0.68). The overallsatellite; namely, CALIOP Level 2 Aerosol Layer Product, was veryFeature Detection AlgorithmsPBL height. Lidar measurements revealed that the maximum PBL height was reached approximately three hours aftersolar noon, whilst the daily evolution of the PBL was completed, on average, one hour earlier. The PBL diurnalwas also analyzed using ECMWF estimations, which produced a stronger cycle during the winter and pre-monsoonThe seasonal analysis of~~

[revised manuscript text omitted]
 1).In 2008, the highest temperature was recorded in May, with a monthly maximum temperature of 36.9 °C. The annual mean temperature was 24.6 °C in 2008 and 25.4 °C in 2009. Monthly maximum temperatures during May and June were 3 to 4 °C lower than the climatological values (World Meteorological Organization, http://worldweather.wmo.int/en/city.html?cityId=224), while February and March (temperature average of March 2008 and March 2009) were characterized by almost 3 °C higher maximum temperatures, as shown in Figure 1(a). The year 2008 exhibited the most rainfall, between June and September, compared to the four year period 2006-2009, with a total of 570 mm in Gual Pahari (Hyvärinen et al., 2010). However, rainfall (June September) was lower than the climatological value of 602 mm in New Delhi. In the monthly periods from April to June and August to September 2008, the total precipitation was higher than the one expected from climatology, with a maximum anomaly appearing in May, whereas in July 2008 cumulative precipitation was lower (Figure 1(b)). This year also exhibited an early monsoon onset date on 16 June, which was one of the earliest onset dates recorded in the area with rainfall data available since 1901 (Tyagi et al., 2009). The Indian summer monsoon in 2008 was somewhat weaker than normal, following the La Niña condition in the tropical Pacific (Lau et al., 2009).

**3 Methodology and Instrumentation3 Methods**

[revised manuscript text omitted]
. TheHence, the PBL height could be identified in 53% of the cases with sufficient data availability (72 days). In Figure 2, the data coverage is presented on a monthly basis in Figure 2. The highest PBLHPBL detection frequency was achieved in February, whichreaching 74%. This high detection rate can be attributed to favorable meteorological conditions, since in February the occurrence of low clouds appeared sparsely without anywas 0.7% with no rainfall events.

**3.2 Radiosonde measurements**

The Bulk Richardson Number (BRN) method was used for PBL determination, employing the formula introduced by Menut et al. (1999):

$$Ri_b\,(h) = \frac{g(h - h_0)}{\theta(h)}\,\frac{[\theta(h) - \theta(h_0)]}{u(h)^2\,+\,v\,(h)^2}$$

, where h is altitude, $h_0$ the altitude of the ground, g gravitational acceleration, θ potential temperature in Kelvin and u and v the zonal and meridional wind components, respectively. The PBL height was determined to be the lowest altitude where BRN reached the critical value, which is taken equal to 0.21 (Vogelezang 1996). Beyond this critical value of $R_i$, the atmosphere can be considered stable and fully decoupled from the underneath layer.

**3.23 Space-borne lidar observations**

Cloud-Aerosol Lidar and Infrared Pathfinder Satellite Observation (CALIPSO) is an Earth Science observation mission that was launched on 28 April 2006. The vertical resolution of the CALIOP (Cloud-Aerosol Lidar with Orthogonal Polarization) system is 30 m. CALIPSO Level 2 aerosol layer product, provides a description of the aerosol layers, including their top and bottom height, identified by automated algorithms applied in the Level 1 data. Detailed description of the aforementioned algorithms can be found in Vaughan et al. (2004) and Winker (2006). In this study, CALIOP Version V4-10 dataset was used. Currently, no operational CALIOP PBL product is available.

More specificallyIn this study, we applied theuse CALIOP Level 2 Aerosol Layer Product, which provides information on the base and top heights of existing aerosol layers, reported at a uniform 5 km horizontal resolution. Leventidou et al. (2013) evaluated the daytime PBLHPBL height derived by Level 2 Aerosol Layer products over Thessaloniki, Greece, for a 5--year period, making the assumption that the lowest aerosol layer top can be considered as the PBLHPBL height. The aforementioned method was also applied over South Africa, revealing high agreement with ground based observations (Kohronen et al., 20142013). During the measurement campaign, PBLHthe PBL height was also accessed by the space-borne lidar CALIOP, within two overpass distances of, 20 and 101 km from Gual Pahari.

**3.3 WRF4 Atmospheric Modelmodel estimations**

**3.4.1 The ECMWF model**

The WRF model, Version 3.9.1 (Skamarock et al., 2008) was also
order to determine the PBLHPBL height. The simulation domain was
the lidar station in Gual Pahari and three domains with a respective
horizontal resolution of 18 km, 6 km and 2 km were used, where the
two inner domains are two-way nested to their parent domain. The
third inner-most domain covers an area between 75.84-78.46° E and
27.38-29.52°52 ° N. The output is provided every hour. On the vertical
axis, 37 full sigma levels resolve the atmosphere up to 50 hPa (≈ 20
km AGL), with a finer grid spacing near the surface.

University scheme (YSU) (Hong et al., 2006) in conjunction with the land surface model Noah (Chen
and Dundhia, 2001) was used for the estimation of PBL height. In addition, the Rapid Radiative

Transfer Model (RRTM) scheme (Mlawer et al., 1997) for longwave radiation and the scheme of
Dundhia (1989) for shortwave radiation were applied. A surface-layer scheme based on the revised
MM5 similarity theory (Jimenez et al., 2012) as well as the Kain and Fritsch (1990, 1993) scheme for
cumulus parameterization were used.  For microphysics, the scheme proposed by Thompson et al.
(2008) was considered. Regarding land use and soil types, the predefined datasets of Moderate

Resolution Imaging Spectroradiometer (MODIS) with 21 land use classes were used. The initial and
lateral boundary conditions were derived from the National Center for Environmental Prediction
(NCEP) operational Global Fine Analysis (GFS) with 1 ° x 1 ° spatial resolution and were updated
every 6 h. The Sea Surface Temperature (SST) was obtained from High Resolution Real-Time Global
SST (RTG SST HR), with spatial resolution 0.083 ° x 0.083 ° which was renewed every 24 h.

In the YSU scheme, the top of the PBL height under unstable conditions is determined as the first
neutral level based on the Bulk Richardson Number (Ri)BRN calculated between the lowest model
level and the levels above (Hong et al., 2006; Shin and Hong, 2011). Under stable conditions, the
RiBRN is set as a constant value of 0.25 over land, enhancing mixing in the stable boundary layer
(Hong and Kim 2008), whereas it is a function of the surface Rossby number over the oceans, following the study of Vickers and Mahrt (2003). More specifically, the revised Stable Boundary Layer
(SBL) scheme (Hong 2010) computes the exchange coefficients with a parabolic function with height,
as in the mixed layer, in which the top of the SBL is determined by the Ri (Vickers and Mahrt 2004).
This leads to a gradual-and not abrupt-collapse of the mixed layer after the sunset, due to the residual
superadiabatic layer near the surface even in the presence of negative surface buoyancy flux.

Within the frame of three case studies, the default simulated PBLHPBL height from WRF was used to
justify the lidar PBLHPBL heights. Furthermore, WRF profiles of temperature (T), relative humidity
(RH) and potential temperature (θ) were compared to corresponding radiosonde profiles. The
comparison was performed through specific criteria following the guidelines given by Seidel et al.
(2010). In the T criterion, the base of an elevated temperature inversion is considered as the PBL top.

Inversions do not appear in every profile, but when present, their base serves as a cap to the mixing
processes below. In θ profiles, the level of the maximum vertical gradient (Oke, 1988; Stull, 1988;
Sorbjan 1989; Garrat, 1992) was used, since this gradient is indicative of a transition from a

convectively less stable region below to a more stable region above. In a similar way, the level where

**4 Results and Discussiondiscussion**

**4.1 Applicability of the WCT method: Case studies**

It was found that in some cases the presence of multiple aerosol layers and low clouds can pose difficulties in PBLH detection (Section 3.1.3). However, these difficulties can be dealt with the use of proper WCT threshold and cut off values (Section 3.1.2). Three case studies of PBLHPBL daily evolution were analyzed and evaluation with ancillary data sources was performed so as to investigate capabilitiestheir strengths and limitationslimits. First, the evolution of PBLHPBL under cloudless conditions is discussed for 12 February 2009. Subsequently, a two-day case with a multiple aerosol layer structure is presented for 1-2 March 2009. Finally, the diurnal development of PBLHPBL is investigated in the presence of low clouds for 29 June 2008. The three criteria (T_crit, RH_crit, θ_crit) were used to determine PBL height in each radiosonde profile. These criteria were also applied to WRF It was found that the presence of multiple aerosol layers and low clouds can pose difficulties in PBL top detection (Section 3.1.3). However, as it will be shown these difficulties can be dealt with the use of proper WCT threshold and cut off values (see Section 3.1.2).

The diurnal evolution of PBL during 12 February 2009 was characterized by an almost constant daily growth rate (133 m/h between 06:00 UTC and10:00 UTC) with a maximum height of 950 m (is presented in Figure 3). Sunrise was approximately at 01:30 UTC, while sunset was at 12:40 UTC. No aerosol layers were observed in the free troposphere. Althoughfound aloft. Between 06:00-12:00 UTC, although internal gradients (yellow and red color) of aerosol content, appeared inside the PBL (06:00-12:00 UTC), the default signal decrease threshold (0.05) was efficient. However, later (Between 12:00-18:00 UTC) in order to avoid strong gradients in the lower parts of the PBL, higher a threshold (of 0.08) was used in conjunction with a 90 m cut-off heights (90 m). Furthermore, height. Due to low aerosol load conditions were responsible for high variability incontent, the derived PBLH (PBL heights between 12:00-14:00 UTC). showed high variability. An almost constant daily growth rate of 133 m/h was found from 06:00 UTC to 10:00 UTC. The maximum height of 950 m was reached at 10:30 UTC. During convective hours (05:00-12:00 UTC), WRF overestimated the PBLH mainly due to the simulated neutral profile-virtual potential temperature at the surface similar to that around 1100 m AGL (differences < 0.5 K, not presented), resulting in an increase in the PBLH (Kim et al., 2013). It is worth mentioning that during the convective period FMI-Polly[XT] identified a light aerosol load activity at the altitude where the numerical models estimated the PBLH, with the WCT technique not detecting this activity due to the weakness of the aerosol gradients. During night-time model estimations yielded lower PBLH compared to lidar data.. During night-time model estimations yielded lower PBLH compared to lidar data. The low wind field produced by the WRF close to the surface (wind speed values up to 3 m/s in the first kilometer) and, thus, the lack of sufficient mechanical turbulence, can be related to the shallow nocturnal PBL. It should be noted that the measured PBLH is expected to depict, apart from any mechanically-driven layer during the stable and transition periods, the top of the previous day's residual aerosol layer, while the simulated PBLH from WRF refers to the height of the shallow mixed layer. Therefore, their difference is expected since they depict different layers. The overall correlation was satisfying (r=0.8).

Figure 3 (middle panel) shows the development of PBL according to FMI Polly^XT measurements and the estimations from the two atmospheric models. During convective hours (05:00-12:00 UTC), the WRF and ECMWF models seemed to overestimate PBL height. On the other hand, during night-time model estimations yielded lower PBL heights compared to lidar data, since the former estimated the nocturnal PBL while the latter identified the RL top. Between 6:00 and 12:00 UTC, FMI Polly^XT identified a light aerosol load activity at the altitude where the WRF model estimated the PBL height. The correlation with lidar hourly heights was satisfying (r=0.8) for both model output data, while a

During the two-day period of 1-2 March 2009, a complex aerosol layer structure appeared in the free troposphere up to 3 km altitude (Figure 4). However, appropriate modification ofthe modified WCT method managed to detect the top of the PBL in most of the 15 min intervals, after modifying the signal decrease threshold and use of appropriate applying a cut-off heights allowed for the detection of PBLH. In order to avoid gradients in the lower parts of the PBL, the signalheight. The threshold was adjusted (within the range of 0.03-0.08) within , which corresponds to a 6-16% signal decrease, in combination withrespectively. Furthermore, a 30-60 m cut-off zoneheight was used, in order to avoid gradients in the lower parts of the PBL.

On 1 March 2009, the transition period (02:00 to 05:00 UTC) was characterized by a slow PBLHPBL development (of 14 m/h), whereas the PBLHPBL evolution was more pronounced in the convective period (05:00 to 09:00 UTC) with a mean growth rate of 101 m/h. The maximum height (of 950 m) appeared at 08:45 UTC. On the next day, a stronger but slightly shorter PBLHPBL cycle was observed, with a mean evolution rate of 187 m/h, reaching a maximum height (of 1010 m) at 08:15 UTC15UTC. This slight modification in PBLHthe development of the PBL, can be attributed to the combination of higher temperature and lower wind speed conditions duringcharacterizing the second day.

**4.1.3 Case with low clouds: 29 June 2008**

In this case broken cumulus clouds appeared between 600-1100 m (from 00:00 to 12:00 UTC). On average, a moderate PBLH evolution (86 m/h) was found, with a maximum height (1279 m) appearing at 9:15 UTC (Figure 5). Whenever clouds appeared below 1km, we made the. The assumption that the cloud base isconstitutes an approach to the top approximation of the PBL, however top was made. However, it could be argued that the PBLHPBL top was located at a higher level, wheresince diffuse aerosol layers were found. In additionalso present there. During this period, it was difficult to findlocate an adequate signal decrease; the default gradient; a threshold corresponding to 10% decrease was used, while sensitivity tests with thresholds sensitive to weaker gradientslower threshold values yielded the same results. Hence, the algorithm exhibited decreased sensitivity, which can be attributed is mainly related to the existence of diffuse aerosol layers. High PBLH was observed immediately after, due to Large PBL height values also appeared around 12:00 UTC, corresponding to a strong aerosol layer which sprawled to lower heights, either through probably due to dry removal or precipitation that evaporated before reaching the ground. Following a short Rainfall was observed between 13:30 and remaining aerosolaerosols kept being displaced downwardsin the downward direction, creating strong the The effect of aerosol removal effect was clear (can be seen between 16:00 and 24:00 UTC) aerosol load, observed, complicated the detection of PBLH was complicated and, hence, accountedthe PBLH (the detected PBL heights between 16:00 and 24:00 UTC).

WRF and ECMWF estimations correlated well with FMI-PollyXT hourly PBL height data (r=0.74). and r=0.69, respectively). During daytime, WRF slightly overestimated PBLHPBL height, while it should be noted that FMI-PollyXT identified intermittent aerosol gradients atan underestimation was observed during night-time by both models. Good agreement was corroborated by additional statistical parameters. Fractional bias was equal to 0.015 and 0.11 for WRF and ECMWF estimations, respectively.

**4.2 Comparison of lidar PBL heights to ancillary data sources**

During the measurement period, 24 CALIPSO overpasses were available withininside 1º radius around Gual Pahari station. The Boundary Top LocationIn 17 cases the boundary top location algorithm (SIBYL (, Selective Iterated Boundary Locator) identified two to four layers (17 cases), while, whilst in the remaining7 cases no layers were identified. However, For the 17 cases, the PBL top from the ground-based lidar observations were not was available in all for 14 cases. In one case, the top of the cases (only in 14).second layer was chosen, as the first one was inside the PBL, according to the attenuated backscatter image from CALIOP. Furthermore, some5 cases (5) were excluded fromnot included in the comparison as the detected layers were clearlyeither above the typical PBL limits (higher than (height >3 km).) or in the free troposphere (height >10 km).

**4.3 Statistical Analysis**

**4.3.1 PBL Diurnal Cycle of PBLH**

Figure 8a shows the mean diurnal PBL evolution as obtained by lidar measurements and ECMWF estimations. Although night-time PBLH isPBL height values were not taken into account for the statistical analysis of PBL seasonal PBLH (Section 4.4.2),height, nocturnal PBLH is consideredvalues are included here in orderso as to investigatepresent the PBL diurnal evolution.

ECMWF estimations revealed a shorter but stronger PBL growth period, with a maximum top height of 2137 ± 143 m, which appeared earlier than the one given by FMI-PollyXT. As in the annual and winter diurnal cycle, ECMWF overestimated PBL top height during convective hours. On the other hand, underestimation was observed during the early morning hours, with a more significant underestimation during night-time due to the fact that FMI-PollyXT identified the RL, whereas the ECMWF estimated the nocturnal PBL top. The total comparison reached an r of 0.84.

In this Sectionthe following, we statistically analyzepresent the main statistical findings regarding the lidar measurements in conjunction with theduring 72 days. The seasonal cycle of mean and maximum seasonal mean PBLHPBL height was found at 695 ± 146 m during winter, (17 days), 878 ± 297 m period (15 days) and 1025 ± 296 m during the monsoon, The (40 days). Regarding the seasonal at 1191 ± 516 m during winter, 1326 ± 565 m during the pre-monsoon period and at 1361 ± 350 m during the monsoon. In general, the PBLH seasonal cycle followed the temperature cycle very well. The temperature cycle of the  measurement days was fairly representative of the whole
seasonal cycle, with the temperature distribution being similar to the distributions of the whole seasonal periods. During the measuring period, a mean temperature of 21 ± 4 ºC was found in  winter, 27 ± 3 in the pre-monsoon and 30 ± 2 ºC in the monsoon season while the seasonal average maximum temperature  was recorded at 29 ± 5 ºC, 33 ± 4 ºC  and 35 ± 2 ºC accordingly. Nevertheless, it should be mentioned that
In winter, the daily mean PBLH distribution was narrower (in majority between 600 and 900 m) compared to the pre-monsoon and monsoon seasons (mostly between 900 and 1200 m). Following a similar pattern, the daily maximum PBLH was rather confined in winter (in majority between 900 and 1200 m) with a significantly broader spectrum (between 600 and 1800 m) in pre-monsoon and monsoon. The highest inter-seasonal variability was exhibited during  pre-monsoon
, which could be attributed to  meteorological conditions. The pre-monsoon season comprised days,  with heavy rainfall and days with hardly any precipitation, which can potentially explain the  broad distribution of daily mean PBLH 251 1191 m). In
winter large  inter-seasonal variability  of attributed to the broad inter-seasonal  temperature range  (20 ºC - 36 ºC).

During the distribution of daily growth rates is presented in Figure 9. For the whole measurement period, daily evolution rates were mostly within  100-200 m/h but lower rates (
29-100 m/h) .  were observed as well. In winter , daily growth rates presented a slightly broad distribution (mostly between 100 and 200 m/h) with (40%), while the pre-monsoon,
slightly, higher growth rates were observed (mainly within 100-300 m/h), with an average of  206 ± 134 m/h. Additionally, rates between 0-100 m/h and 500-600 m/h within the range 100-200 m/h, while 33% were observed, following the pattern of between 200 and 300 pre-monsoon season (Section 4.4.2). In the monsoon, season, lower evolution speeds were slightly lower observed (121 ± 67 m/h), N=22), with 45% being less than 100 m/h., while a significant percentage (40%) was found between 100 and 200 m/h.

The average PBL height was lower in Gual Pahari (866 m ) in comparison to Elandsfontein (1400 m) with less seasonal variability (standard deviation of 165 m in Gual Pahari; 500 m in Elandsfontein). In both sites the maximum PBL height was reached approximately three hours after the solar noon, since the daily solar cycle is similar in the latitudes of the two stations. In Gual Pahari, the highest rates (mostly within 100-300 m/h) appeared in the pre-monsoon season (April-May), whilst in Elandsfontein maximum rates (between 120-320 m/h) were reached during spring. (September-October, (Kohronen et al., 2014) a period that exhibits strong similarities with the ). The pre-monsoon season in India. and the spring season in South Africa have strong similarities.

**56 Summary and Conclusions**

In this studypaper, one year long ground-based lidar measurements were used to retrieve PBLHanalyze PBL height variability over Gual Pahari, New Delhi. The feasibility of deriving PBLH with the modified WCT technique was investigated and the respective resultslidar retrieved PBL heights were compared to data from independent sources.

In; radiosondes, satellite observations and two atmospheric models. Three case studies of PBL daily evolution were discussed so as to identify atmospheric structures which can complicate PBL height detection. It was found, in support of previous work (Baars et al., 2008; Korhonen et al., 2014), it was found2013), that the modified WCT method exhibited satisfying efficiency performed well under different meteorological and aerosol load regimes. On a case with elevated aerosol layers, More specifically, a significantly good performance was revealed, even when the layers were injected into the PBL. Such layers have been reported in literature as on a major challenge in the attribution of the PBLH especially during night-time (Haeffelin et al., 2012). PBLHtwo day case, with multiple aerosol layers aloft. However, PBL determination was complicated in the presence of before a rain event, where lofted layers created strong aerosol content gradients and later on, where diffuse aerosol layers.

Low aerosol load, observed mainly during morning or afternoon transitions, also represents a condition for uncertain determination of PBLH (Haeffelin et al., 2012). Sensitivity analysis revealed stable performance of the WCT algorithm, with the exception of elevated layers and PBL internal gradients, which affected the results when specific thresholds were applied. Higher thresholds appeared to be more sensitive towards detecting lofted layers.

In the context of the aforementioned cases, WRF model case studies, numerical estimations overestimated PBLHPBL height in the daytime, while an underestimation was observed in the night-time. The understanding of turbulence in nocturnal SBL and its parameterization is rather slow and not well established in NWP models (Mahrt et al., 1999; Beare et al., 2006; Hong 2010). In this study, this is  partly, addressed by the revised SBL scheme that retains the turbulent levels so as to avoid the abrupt collapse of the mixed layer after the sunset by using the exchange coefficients. ~~attributed to the fact that the lidar system identified the RL, whereas the numerical models estimated the nocturnal PBL top. The comparison between radiosonde and WRF vertical profiles, through three different methods, showed that radiosonde data overestimated PBL height in the night-time. The discrepancies between radiosonde and WRF PBL heights could be attributed to various sources, such as the different vertical resolution and the different nature of each data set; radiosondes provide in situ measurements, whereas WRF model provides numerical estimations of various meteorological parameters.were~~are included in the simulations could also explain the model underestimation. Furthermore, it should be noted that the measurements often depict different layers from the simulated ones, as in the case of the residual aerosol layer.

During the rainy season of monsoon, the diurnal cycle of PBLH was weaker and its evolution was completed earlier. A relatively warmer and drier winter and, respectively, a colder and rainier pre-monsoon were observed compared to climatological records. These meteorological patterns could account for the observed PBLH cycle, which was rather indistinct compared to the cycle expected from long-term climate statistics. Daily evolution rates of 29-200 m/h were mainly observed, with lower rates during the rainy season of monsoon.~~The evolution of PBL started two to four hours after sunrise and was completed two hours after the solar noon, with the maximum PBL height observed approximately one hour later. In the winter and pre-monsoon season, ECMWF data revealed a stronger PBL daily evolution. During the monsoon season, both FMI Polly[XT] measurements and ECMWF output data, produced a smoother diurnal cycle, consisting of weaker fluctuations between daytime and night time, with PBL heights from ECMWF being systematically lower than those derived from FMI.~~ Future studies are necessary in order to better understand the factors that modulate the exchange of moisture, heat and momentum between the surface and PBL and, consequently, affect the comparison of modelled PBLH with observational data. In addition, the relative contribution of the various PBL dynamics drivers, under different aerosol load and meteorological regimes, needs to be further investigated. The feasibility of applying the modified WCT method in simpler lidar systems such as ceilometer and Doppler lidar, should be assessed. These systems entail less operational cost and, thus, exhibit good potential for determining the PBLH and evaluating weather prediction and pollution dispersion models on an operational basis. In recent years, significant effort has been made towards the establishment of ceilometer networks by national weather services and other agencies over Europe with the aim to build up a framework for real-time applications and improvements of air quality and weather prediction by assimilation of ceilometer data (Haeffelin et al., 2012; Wiegner et al., 2014). Analogous efforts are currently in progress over different parts of India, like in the states of Maharashtra and Kerala and in the union territory of Delhi (Sharma et al., 2016; Babu et al., 2017; https://www.lufft.com/projects/several-lufft-chm-15k-ceilometer-projects-in-india-529/).

mean and maximum PBL height appeared in the monsoon season, where the highest mean and

**Appendix A: Sensitivity Analysis of the WCT threshold**

[revised manuscript text omitted]

**Figure 1:** Maximum temperature and cumulativetotal precipitation during the measurement campaign
(black) and anomalies (blue)anomaly at New Delhi on a monthly basis. Anomalies representThe bars
indicate the difference between the climatological values and the corresponding values during the
measurement campaign. Climatological values were obtained from World Meteorological Organization
( http://worldweather.wmo.int/en/city.html?cityId=224) for the site of Safdarjung airport.

[Figure]

**Figure 2:** Data coverage of lidar measurements  calculated with respect to total convective hours (from 4 h after sunrise to 1 h before sunset) during the measurement days of the campaign.

[Figure]

[Figure]

[Figure]

[Figure]

**Formatted Table**

[Figure]

**Figure 3:** Evolution of PBLH observed on 12 February 2009. Range-corrected signal (top) at 1064 nm as measured with FMI-Polly$^{XT}$. Black  lines indicate 15-min PBLH, while black zones in the lower part of the figure indicate the extent of the signal cut-off area. The colorscale is normalized on a 6-hour basis, with red and yellow indicating higher aerosol load, while green and blue lower load, respectively. PBLH (bottom) as  given by the FMI-Polly$^{XT}$ and WRF (vertical lines indicate sunrise and sunset times).

[Figure]

[Figure]

**Figure 4:** Same as Fig. 3 except for 1-2 March 2009. White horizontal lines (top) indicate 15-–min values of cloud base height.

[Figure]

Formatted Table

**Figure 5:** Same as Fig. 3 except for 29 June 2008.  Grey shading (middle) indicates rainfall.

[Figure]

**Figure 6:** XT

[Figure]

**Figure 6:** PBLH comparison  for FMI-Polly^XT and CALIOP . The heights given by CALIOP have been corrected with elevation. The markersize is proportional to the overpass distance from the ground-based lidar.

[Figure]

[Figure]

**Figure 89:** Frequency distribution of daily mean (green) PBLH, daily and maximum PBLH (red) and daily mean growth rate (blue)PBL height as calculated duringthroughout the measurement period (a), the winter period (ab), the pre-monsoon season (be) and the monsoon period (cd). Numbers indicate data availability.

[Figure]

**Figure 9:** Sensitivity analysis of the WCT method for the case of 2 March 2009. PBLH was estimated by FMI-Polly[XT], after modification of the WCT threshold, and by WRF model.

---

## Author Response (AR2)

We would like to thank the two reviewers for devoting time to read our manuscript and provide valuable comments for improving it and increasing its scientific value. We have modified our manuscript following the guidelines given by the two reviewers. Below we answer to each reviewer's comment (RC) separately. The **RC**s are given in **bold**, our replies in plain font and the corresponding *changes in the manuscript* are given in *italic*.

Kind regards,

Konstantina Nakoudi, on behalf of all the co-authors

Comments by Anonymous Referee #2 and our replies and changes in manuscript:

We would like to thank the Anonymous Referee 2 for the constructive comments and recommendations. We made strong efforts to revise the manuscript. More emphasis has been given on the reasons behind the arisen discrepancies between the different methodologies. Comparisons were directed between ground-based and space-borne lidar measurements as well as numerical estimations from WRF. The comparison between lidar and radiosondes is not included in the new version of the manuscript, since the nature of the two methods is different. Furthermore, we decided to include one atmospheric model in the discussion and we selected the WRF, which has higher horizontal resolution than ECMWF. The comparison between WRF and radiosonde profiles and the application of the temperature, potential temperature and relative humidity criteria is excluded as well, since the comparison with lidar is indirect. As suggested by the two reviewers, Section 5 (Comparison to another location) was left out. A new Section (4.2) has been added, in which the sensitivity analysis regarding the WCT threshold is discussed.

1) Lines 45-47, I suggest adding more details about the detection of PBL height using radiosonde measurements and other instruments (e.g. microwave radiometers) with the appropriate references.

More details have been added regarding PBL height detection from radiosonde measurements and other instruments such as microwave radiometer and Doppler wind lidar together with the respective references.

The corresponding part in Section 1 (Introduction) has been modified as follows:

Several methods have been proposed to estimate PBLH, utilizing vertically resolved thermodynamic variables, turbulence-related parameters and concentrations of tracers (Seibert et al., 2000; Emeis et al., 2004). More specifically, different methods for the determination of the PBLH from radiosonde measurements have been compared and the associated uncertainties have been estimated (Seidel et al., 2010; Wang and Wang, 2014). Radiosondes have been routinely used for decades and therefore are a valuable method for long-term climatology analyses (Seidel et al., 2010; Wang and Wang, 2014). Restrictions of radiosondes refer to the coarse vertical resolution of standard meteorological data with respect to boundary layer studies as well as the smoothing due to the sensor lag constant bounded by the high ascent rate of the radiosonde (Seibert et al., 2000). Remote sensing systems such as aerosol lidar, microwave radiometer (Cimini et al., 2013), wind-profiling radar (Cohn and Angevine, 2000) and Doppler wind lidar (de Arruda Moreira et al., 2018) are suitable for long-term measurements of various atmospheric quantities with high temporal resolution and they can be used either independently or synergistically so as to retrieve the PBLH. Space-borne lidar systems provide the advantage of spatial coverage,

although for studies focusing on a specific area of interest, measurements are constrained by the overpass frequency. Ceilometers entail less cost, but on the other hand, they include fewer channels and, thus, cannot be used for detailed aerosol studies. In elastic and Raman lidar systems atmospheric aerosols are used as tracers and the PBL top is indicated by a gradient in the range-corrected lidar signal (Menut et al., 1999; Brooks 2003; Amiridis et al., 2007; Morille et al., 2007; Baars et al., 2008; Engelmann et al., 2008; Groß et al., 2011; Tsaknakis et al., 2011; Haeffelin et al., 2012; Scarino et al., 2013; Summa et al., 2013; Korhonen et al., 2014; Lange et al., 2014; Bravo-Aranda et al., 2016).

2) Lines 47-54, not all the lidar systems can provide continuous, systematic and quantitative measurements of atmospheric aerosol profiles. Please clarify this aspect and expand this section, adding strengths and weaknesses between space-borne and ground-based systems and between lidar ceilometers and research lidars.

We thank the reviewer for this remark. We included information about the advantages and disadvantages of satellite lidar systems and ceilometers:

Space-borne lidar systems provide the advantage of spatial coverage, although for studies focusing on a specific area of interest, measurements are constrained by the overpass frequency. Ceilometers entail less cost.

Furthermore, we replaced the sentence Lidar (Light Detection And Ranging) systems can provide continuous measurements ... with the sentence:

Remote sensing systems such as aerosol lidar, microwave radiometer (Cimini et al., 2013), wind-profiling radar (Cohn and Angevine, 2000) and Doppler wind lidar (de Arruda Moreira et al., 2018) are suitable for long-term measurements of various atmospheric quantities with high temporal resolution and they can be used either independently or synergistically so as to retrieve the PBLH.

3) Lines 93-104, to understand if the anomalies in temperature and precipitation are significant, you should also report in the discussion and in figure 1, the standard deviation of the measured and climatological values.

The standard deviation of the climatological values is not available. The World Meteorological Organization provides only the climatological value for the mean daily maximum temperature and mean total rainfall, which can be accessed at http://worldweather.wmo.int/en/city.html?cityId=224. Standard deviations of the average maximum temperature and cumulative precipitation are discussed in the new Section 4.4.2 and their relation to the seasonal PBLH cycle is investigated.

4) Lines 121-122, please specify the differences between nighttime and daytime configurations and the corresponding vertical sounding ranges.

The configuration of the lidar system FMI-PollyXT was the same during daytime and nighttime. The lower limit of the vertical sounding range was depended on the height, where full overlap between the emitted laser beam and the receiver field of view was achieved. During the measurement campaign, the altitude of full overlap varied from 550 to 850 m. Vertical range covers the whole troposphere under cloudless conditions. This is sufficient for PBL studies considering the heights needed in the study. Engelmann et al. (2016) reports a maximum vertical range of 40 km, which depends on the capabilities (height bins) of the data acquisition.

These aspects are discussed in Section 3.1.1 of the manuscript:

The system vertical resolution was 30 m and the vertical range covered the whole troposphere under cloudless conditions. This is sufficient for PBL studies considering the heights needed in the study. Engelmann et al. (2016) reports a maximum vertical range of 40 km, which depends on the capabilities (height bins) of the data acquisition. The FMI-PollyXT lidar system is described in more detail in Althausen et al. (2009) and Engelmann et al. (2016).

The incomplete overlap between the laser beam and the receiver field of view L-R (Laser-Receiver), restricted the observational detection range to heights above 200-300 m. This was partly counterbalanced by the overlap correction function. In this study, overlap corrections were performed at 532 nm following the methodology proposed by Wandinger and Ansmann (2002). During the measurement campaign, the L-R was completed at 550-850 m.

5) Lines 125-127, please add more details about the overlap factor of the system. Is a correction applied? Which is the height of full overlap? Since incomplete overlapping could hamper the PBL height detection, this characteristic should be well specified.

In this study, a correction was used for incomplete overlap following the methodology proposed by Wandinger and Ansmann (2002). Full overlap between the emitted laser beam and the receiver field of view was achieved between 550 and 850 m.

These aspects are discussed in Section 3.1.1 of the manuscript: *The incomplete overlap between the laser beam and the receiver field of view L-R (Laser-Receiver), restricted the observational detection range to heights above 200-300 m. This was partly counterbalanced by the overlap correction function. In this study, overlap corrections were performed at 532 nm following the methodology proposed by Wandinger and Ansmann (2002). During the measurement campaign, the L-R overlap was completed at 550-850 m.*

- 6) Lines 143-145, and 151-154, see the comment for lines 125-127. The overlap characterization should be discussed in details in section 3.1.1. As the reviewer suggests, we have included more information concerning the overlap characterization in Section 3.1.1. Please see our replies in Comments 4 and 5.
- 7) Lines 148-150, the WCT method was also applied for the detection of cirrus clouds height base (Dionisi et al., 2013, ACP) where a sensitivity study was made to fix a proper threshold. Please add this reference.

We thank the reviewer for letting us know about the study of Dionisi et al. (2013). The reference to this study has now been added to the manuscript as follows:

The WCT method has also been applied for the detection of cirrus cloud base height (Dionisi et al., 2013, Voudouri et al., 2018) over different geographical regions.

**8) Line 220, please specify the CALIOP version dataset used in this study.**

In this study, CALIPSO Level 2 aerosol layer, Version V4-10, product was used.

This information has been added in in the corresponding Section of the manuscript (now 3.2, previously 3.3):

In this study, CALIOP Version V4-10 dataset was used. Currently, no operational CALIOP PBL product is available.

**9) Lines 225-227, please specify if the used CALIOP overpasses are nighttime or daytime measurements.**

The cited study of Leventidou et al. (2013) used daytime lidar measurements. This is now specified in the corresponding Section of the manuscript (now 3.2, previously 3.3):

Leventidou et al. (2013) evaluated the daytime PBLH derived by Level 2 Aerosol Layer products over Thessaloniki, Greece, for a 5-year period, making the assumption that the lowest aerosol layer top can be considered as the PBLH.

10) Please specify the impact of the WCT threshold on the results. A sensitivity study could be of help to interpret the results. Which is the associated error to the estimated PBL height? Is there any correlation between the magnitude of this error and the agreement between the different datasets? The effect of the different model horizontal and vertical resolution should be also added in the discussion.

We would like to thank the reviewer for this recommendation. As suggested, we performed a sensitivity analysis regarding the WCT threshold impact on the PBLH values. For this reason, we used the case of 2 March 2009, which was already discussed in the applicability of the WCT Section (4.1). The threshold was modified from 0.03 to 0.08, which corresponds to 6-16% signal gradients. It was found that the overall performance of the WCT algorithm is stable, with the exception of elevated layers and strong gradients appearing inside the PBL. Subsequently, it has been observed that the agreement with the simulated PBLH from WRF was affected in the presence of elevated layers or internal aerosol content gradients. However, this deviation was dependent on the altitude of the aforementioned atmospheric features. Furthermore, during early morning, where the convective activity has not been initiated yet, a small fluctuation (30 m) was identified.

A new Section (Section 4.2) has been added in manuscript, discussing the sensitivity analysis as follows:

In cases of elevated layers or aerosol gradients within the PBL, it has been revealed that the signal decrease threshold needs to be properly adjusted (Section 4.1). In this study, we adapted the threshold (t) so that the WCT algorithm was allowed to identify signal gradients in the order of 6-16% (t=0.03-0.08, correspondingly). In this Section, we investigate the effect of the WCT threshold on the estimated PBLH. For this reason, we performed a sensitivity analysis modifying the signal decrease threshold for the case of 2 March 2009, where elevated layers were injected into the PBL.

The overall performance of the WCT technique was stable (Figure 6), with the threshold affecting the results in only a few cases. When the lowest and more sensitive to detect weak layers threshold (0.03) was applied, a thin aerosol layer (around 1300 m) was identified (see Figure 4). At this time, increased thresholds (0.04-0.08) detected a stronger elevated layer (approximately at 2 km). The lowest threshold was also more efficient when gradients appeared inside the PBL (around 17:00 UTC), with the higher thresholds yielding increased PBLH by approximately 300 m. When the elevated layers were characterized by higher aerosol load (18:00-19:00 UTC), lower thresholds (0.03-0.05) performed better as well, with the higher ones identifying stronger layers (around 1 km). Thus, the PBLH deviation, introduced by the modification of the WCT threshold, appeared to depend on the altitude of internal gradients or elevated layers. However, in the early

morning (00:00-03:00 UTC), where the convective activity was not initiated yet, a minor fluctuation (30 m) was observed, related to the algorithm's sensitivity towards aerosol content gradients.

An adequate threshold adaptation also affected the agreement with the modelled PBLH. More specifically, it is shown (Figure 6) that during cases where the applied threshold induced a deviation from the smooth PBLH evolution, the disagreement with modelled PBLH increased as well. Besides, the agreement with the simulated PBLH appears to depend on the altitude of the atmospheric features (internal or elevated aerosol gradients) that affect the performance of the WCT algorithm.

**Figure 6:** Sensitivity analysis of the WCT method for the case of 2 March 2009. PBLH was estimated by FMI-PollyXT, after modification of the WCT threshold, and by WRF model.

**11) Lines 313-317. Please explain the difference on PBL heights retrieved by radiosonde and WRF model.**

The differences between the PBL heights derived by radiosonde and WRF model can be possibly attributed to the different vertical resolution of the radiosonde and WRF profiles. In the revised manuscript, we decided to exclude the WRF-radiosonde profile comparison because there was no direct comparison with lidar.

12) Lines 384-386. The results of the PBL height comparison between lidar and rds during daytime seems due to the sum of two opposing effects: the overestimation and underestimation of PBL height by rds, respectively. In fact, few points are along the 1:1 black line of figure 6, with two clouds of points on the right and on the left of the 1:1 line. This is confirmed by a significant but not very satisfying correlation (R2 = 0.46). Please explain this effect and rephrase this section. Please add the statistical significance of the comparison.

As the reviewer points out, in the scatter plot for comparing lidar to radiosonde PBLH there are data points on the left and right side of the 1:1 line. This effect can be explained by the different vertical resolution of lidar and radiosonde measurements. Furthermore, the different methodologies for the determination of the PBL height can account for discrepancies as well as the distance between the lidar station and the radiosonde launch site. The correlation coefficient for daytime measurements was found 0.68.

As the reviewer suggests, we performed a statistical significance test. More specifically, if p-value is lower than the significance level (0.05 in this case), then the corresponding correlation between the two datasets is considered significant. In other words, there is 5% probability that there is no relationship between the two datasets (null hypothesis).

The p-value for the correlation between lidar and daytime radiosondes was found equal to 0, while for lidar and night-time radiosondes was equal to 0.03. Since, the p-values are lower than the statistical significant level of 0.05, the correlation between the two datasets is considered significant.

In the revised version, we do not discuss the lidar-radiosonde comparison due to the different nature of the two methodologies. Therefore, previous Sections 3.2 and 4.2.1 as well as previous Figure 6 have been removed.

**13) Lines 405-409. Please add the statistical significance of the comparison.**

As the reviewer suggests, we performed a statistical significance test as in Section 4.2.1. We found that for ground-based and satellite lidar the p-value was equal to 0.005, which is lower than the statistical significant level of 0.05. Hence, the correlation between the two datasets is considered significant at the 95% confidence level. In other words, there is 5% probability that the null hypothesis (no correlation between the two datasets) is true.

In the revised manuscript we also comment that longer measurement periods or more extended comparison to ground stations are needed in order to perform robust comparison between ground-based and satellite lidar measurements.

**14) Lines 409-413. The number of considered cases is probably too small to generalize these results. Is there any noticeable difference between the different aerosol type of detected layers or between nighttime and daytime comparisons?**

Based on the analyzed CALIPSO observations, in the majority of cases the detected Aerosol Subtype was dust, with a few cases comprising dust-polluted dust and dust-polluted smoke mixtures. Furthermore, the layer top altitude did not appear to change systematically between daytime and night-time.

The relevant information has now been included in the manuscript:

Based on the analyzed cases, it was found that the overpass distance (here 20 and 101 km) from the lidar station and time difference between the measurements did not affect the agreement of the PBLH. Furthermore, the layer top altitude did not appear to change systematically between daytime and night-time. However, the small number of measurements does not allow us to generalize these findings. Hence, longer measurement periods or more extended comparison to ground stations are needed in order to draw more robust conclusions.

**15) Is it possible to add in this analysis the mean diurnal PBL evolution estimated through WRF simulations?**

The simulation of the diurnal PBLH evolution by WRF was dedicated to a specific number of cases, which are presented in Section 4.1 in order to justify the PBLH derived by the WCT method under different aerosol load and meteorological conditions.

**16) Is there an explanation for the ECMWF overestimation (Polly underestimation) of PBL top height during convective hours for Winter and Pre-monsoon seasons and the ECMWF underestimation (Polly overestimation) for Monsoon season? The good agreement found at 12 UTC should also be highlighted.**

As the reviewer points out, during convective hours in the winter and pre-monsoon, ECMWF overestimated PBLH, while in the monsoon season an underestimation was observed. During the monsoon period high amounts of precipitation are expected, whereas in winter and pre-monsoon much lower amounts are expected. This opposite behavior can possibly pertain to the modelled amount and initiation time of precipitation and subsequent evaporation. In addition, the soil and vegetation parameterization schemes significantly affect the energy and moisture fluxes inside PBL, which depend on the thermal properties of the underlying surface such as heat capacity and heat conductivity. In particular, the phase of soil water plays a key role in latent heat fluxes. It has been suggested that a non-proper representation of water soil phase can lead to a delay in soil cooling in the beginning of the cold period and a corresponding delay in soil warming in spring, an effect which is more intense if the solar forcing is significant as in the subtropical region of Gual Pahari. Both effects make soil temperature less responsive to the atmospheric forcing (ECMWF, 2010b, p.119), and, thus, can possibly account for the seasonal patterns appearing in the PBLH diurnal cycle.

Therefore, the partition between latent and sensible heat fluxes by the surface parameterization scheme of the ECMWF model could explain the reversed behavior during rainy and relatively drier periods.

During times of maximal insolation (6:00 and 9:00 UTC) the overestimation of PBLH by ECMWF was higher, especially in the winter and pre-monsoon seasons, where cloud cover is in general lower. However, in the presence of lower solar irradiance (12:00 UTC) and, thus, weaker thermal turbulence, FMI-PollyXT and ECMWF exhibited the highest agreement, particularly in the winter and pre-monsoon periods, where the solar radiation is expected to be the main driver in the formulation of PBLH. Hence, the good agreement at 12 UTC is most likely related to the intensity of solar irradiance. On the other hand, during the monsoon period, the performance of ECMWF comparison is fairly the same during all convective times (6:00, 9:00 and 12:00 UTC). This can be attributed to the fact that more and more complex factors, such as cloud cover and precipitation, arise and contribute to PBLH development during the rainy monsoon season.

In the revised manuscript, we decided to exclude results from ECMWF due to its low horizontal resolution.

17) The considered cases for this analysis are only 44 whereas for the previous section the number is higher (72). Please explain this difference. Statistical significance should also be specified. The measured differences in the growth rates between premonsoon

**and monsoon season can be attributed to a real signal or the poor significance of the sample does not allow any physical explanation? Please clarify these aspects.**

Following the guidelines given by Baars et al. (2008), PBL growth period began when the PBL height started to increase (typically 2-4 h after sunrise) and was complete when 90% of the daily maximum height was reached (typically between 08:00 and 10:30 UTC). Regarding the daily evolution rate, this was determined through the slope of a linear fit to the hourly height values (between the start and the completion of the growth period). Furthermore, the calculation of the evolution rate was restricted to cases where at least 4 consecutive or 3 non- consecutive hourly values were available. Due to these criteria, the number of the mean growth rates data used in the analysis is lower than the number of the mean and maximum PBLH data.

The two-sided Wilcoxon rank sum test has been applied in order to examine whether the samples of pre-monsoon and monsoon growth rates are statistically different. The test has yielded that the two samples are statistically different at the 95% significance level (p-value=0.03). The differences in the growth rates between pre-monsoon and monsoon could be explained physically. More specifically, the slightly lower growth rates that were observed during monsoon season can be related to the weaker diurnal PBLH cycle that was found during this season. The above mentioned behaviour can be possibly explained by the differences in precipitation between the two seasons. The pre-monsoon season was characterized by less precipitation compared to the rainy period of monsoon. The relevant information has been added in the new manuscript as:

The distributions of daily growth rate during pre-monsoon and monsoon show similarities. In order to examine whether the distributions are statistically different we applied the twosided Wilcoxon rank sum test (Wilcoxon, 1945; Wilcoxon and Wilcox 1964). The test yielded that the two distributions are statistically different at the 95% significance level. Hence, the differences in the growth rates between pre-monsoon and monsoon could be explained physically. More specifically, the slightly lower growth rates observed during monsoon are possibly related to the weaker diurnal PBLH cycle that was found during this season (Figure 8c). The above mentioned behavior can been explained by the different precipitation patterns during the two seasons, since pre-monsoon was characterized by less precipitation compared to the monsoon.

18) I'm not sure that this section is bringing any relevant information. Please motivate this comparison with further details and results or remove the section.

As suggested by the Reviewer, this section has been removed from the manuscript. The comparison with the PBLH characteristics over Elandsfontein site is performed in parallel with the corresponding results (PBLH diurnal and seasonal cycle) from Gual Pahari.

We would like to thank the Anonymous Referee #3 for devoting time in reading and commenting on our manuscript. Following his/her suggestions, the content of the manuscript has been significantly revised. More specifically, comparisons were directed between ground-based and space-borne lidar measurements as well as numerical estimations from WRF. The comparison between lidar and radiosondes is not included in the new version of the manuscript, since the nature of the two methods is different. Furthermore, we decided to include one atmospheric model in the comparison. Therefore, we selected the WRF, which has higher horizontal resolution than ECMWF. The comparison between WRF and radiosonde profiles and the application of the temperature, potential temperature and relative humidity criteria are excluded, since the comparison with lidar is indirect. As suggested by the two reviewers Section 5, (Comparison to another location) was left out. A new Section (4.2) has been added, in which the sensitivity analysis regarding the WCT threshold is discussed. Emphasis is given on the reasons that can explain the discrepancies with ancillary sources. In the same manner, the Conclusions and the Abstract of the manuscript have been rewritten. Excessive details regarding the Figures and Tables have now been removed.

1) P3, L85: Please check the latitude and longitude of the two stations.

The coordinates of the lidar site and the radiosonde launch site have been checked. The radiosonde site is located NE of the lidar station, not NW.

The corresponding text in the manuscript has been now corrected.

2) P3, L93-108: This paragraph presents very detailed information about the meteorological conditions of the observation site, but I think the messages that authors have presented are not well selected and structured...authors could shorten this paragraph and present the information that serves the topics and results that will be present in the following sections.

The paragraph has been reconstructed and shortened as the reviewer suggests. The temperature and precipitation anomalies are used later in the new Section 4.4.2, where the seasonal cycle of PBLH is investigated in relation to the meteorological conditions. The new paragraph is as follows:

Temperature and precipitation patterns can potentially reflect the state of sensible and latent heat fluxes within the PBL as well as the exchange of moisture and momentum with the Earth's surface. Thus, climatologies of meteorological parameters can be considered a valuable tool for assessing the representativeness of PBLH seasonal cycle with respect to long-term measurements. Such a comparison is performed in Section 4.4 based on the 30-year anomalies of maximum temperature and accumulated precipitation (Figure 1).

3) P3-P4: I think the lidar system deserves more 'words', even it has been presented in other papers. Main messages should be addressed to the readers.

More details regarding the technical specifications of the portable Raman lidar system FMI-PollyXT have been added in the manuscript (Section 3.1.1), such as the emitted wavelengths, the beam divergence and the telescope type and field of view:

The measurements were conducted with a six-channel Raman lidar called FMI-PollyXT (Finnish Meteorological Institute - Portable Lidar sYstem eXTedend). The lidar system was entirely remotely controlled via an internet connection, with all the measurements, data transfer and built-in device regulation being performed automatically. The instrument was equipped with an uninterruptible power supply (UPS) and an air conditioning system (A/C) to allow for safe and smooth continuous measurements. A rain sensor was also connected to the roof cover in order to assure a proper shutdown of the instrument during rain.

*FMI-Polly*XT used a Continuum Inline III type laser. The pulse rate of the laser was 20 Hz and it delivered energies of 180, 110 and 60 mJ simultaneously (with external second and third harmonic generators) at three different wavelengths, i.e. 1064, 532, 355 nm, respectively. A beam expander was used so as to enlarge the beam from approximately 6 mm to 45 mm. The remaining beam divergence after expansion was less than 0.2 mrad. The backscattered light was collected by a Newtonian telescope, which had a main mirror with a diameter of 30 cm and a field of view of 1 mrad. The output of the instrument included vertical profiles of the particle backscatter coefficient at three wavelengths, i.e. 355, 532 and 1064 nm (retrieved with the Klett method; Klett (1981) and Klett (1985)), extinction coefficient at 355 and 532 nm (retrieved with the Raman method (Ansmann et al., 1990; Ansmann et al., 1992) by using the Raman shifted lines of N2 at 387 and 607 nm) and linear particle depolarization ratio at 355 nm. The system vertical resolution was 30 m and the vertical range covered the whole troposphere under cloudless conditions.

**4) P4, L145: "A certain range of signal is cut to avoid strong gradients in the lower range", so how do you choose the range to be cut? Is it automatic?**

In cases, where strong signal gradients appeared in the first hundred meters from ground, we made use of the option to cut the lower parts of the signal. This procedure was not automated, but it was performed manually. More specifically, we started cutting the first height bin (30 m) above ground and we tested whether the WCT algorithm managed to omit these strong gradients. If the latter was not successful, then we repeated the above mentioned procedure for the first two height bins (60 m). The algorithm offers the capability to perform the cutting off procedure up to 29 height bins (870 m). In case the algorithm did not manage to detect a significant gradient, then it was not possible to detect the PBL height.

- 5) P4, L152: Comment the abbreviation 'L-R'; and the paper "Wandinger and Ansmann, 2002" is not included in the bibliography. The abbreviation L-R was commented as Laser-Receiver. The publication of Wandinger and Ansmann, 2002 has now been added in the bibliography of the manuscript.
- 6) The Subsection '3.1.3 Data coverage' is very 'dry' and not so interesting. It does provide some information, but it is too detailed with unnecessary technical issues, and some numbers are not well defined so I got confused and sometimes had to redo these simple calculations. I think this paragraph could be shortened.

The Data Coverage Section has now been shortened. An emphasis is given on the factors that either prohibited the operation of the lidar system or hampered the detection of the PBL top. The new Section is as follows:

During the one-year long measurement campaign FMI-PollyXT was measuring on 139 days. Due to technical problems with the laser, the data coverage from September to January was sparse. Furthermore, precipitation prohibited lidar measurements, since the lidar system had to shut down. Hence, sufficient data availability was achieved during 72 days. Multiple aerosol layers appeared mainly between March and May, whereas low clouds were present mostly in the monsoon period and both complicated PBL top detection. Additionally, some technical issues arose due to photomultiplier supersaturation and signal problems. A lack of a significant decrease in the backscatter profile was observed in only a few cases. The latter was a first indication that the modified WCT method can detect the PBL top efficiently, as long as the signal decrease threshold is tuned properly. The data coverage is presented on a monthly basis in Figure 2. The highest PBLH detection frequency was achieved in February, which can be attributed to favorable meteorological conditions, since low clouds appeared sparsely without any rainfall events.

**7) P8, L203: And in Figure 3, Please comment the PBLH close to 0 m, why do the models produce such low values? There are two T profiles in the figure at right, is it a mistake? What are the white squares in Figure 3 upper panel?**

ECMWF produces low PBLH values. The surface layer scheme, which is utilized in the ECMWF model for describing the turbulent transfer of heat, momentum and moisture between the surface and the lower parts of the atmosphere, allows a consistent treatment of different roughness lengths for momentum, heat and moisture. However, it has been found that the revised stability functions reduce diffusion in stable situations producing a shallower stable boundary layer (ECMWF, 2010b, p. 37).

Regarding numerical weather prediction (NWP) models, the understanding of turbulence in nocturnal stable boundary layer (SBL) and its parameterization is rather slow and not well established (Mahrt et al., 1999; Beare et al., 2006; Hong 2010). As a result, there is a tendency of the PBLH to remain at the lowest model level mainly due to the deficiency in SBL mixing and partly due to the poor vertical resolution in NWP models. In particular, the PBLH usually becomes the height of the lowest model level right after sunset. In the present study, this is partly addressed by the revised SBL scheme (Hong 2010) that computes the exchange coefficients with a parabolic function with height as in the mixed layer, in which the top of the SBL is determined by the bulk Richardson number (Ri), following the study of Vickers and Mahrt (2004). This leads to a gradual-and not abrupt-collapse of the mixed layer after the sunset due to the residual superadiabatic layer near the surface even in the presence of negative surface buoyancy flux. However, the fact that neither anthropogenic heat sources nor heat storage in buildings were included in the simulations could also explain the model underestimation during the night.

Furthermore, it should be noted that often the measurements depict different layers from the simulated ones, as in the case of the residual aerosol layer. The comparison could be improved if overall consistency in PBLH retrieval approaches between the model and lidar observations was obtained. The two T profiles were plotted by accident. These profiles, as stated above, are not included in the new manuscript. White squares in the contour plots of the lidar range-corrected signal indicate 15 min values of cloud base height.

8) P9, L311: I miss a clear and quantitative difference between the PBL and RL. In Figure 3, there is a noticeable decrease of lidar signal at 1000-1200 m, do you consider it as the residual layer? Please also check the color scale of Figure 3. I saw a clear

**discontinuity at 06:00 UTC, 12:00 UTC and 17:30 UTC, are the lidar signals ploted with the same color scale? Moreover, the width of the black zone near the surface in Figure 3 is also changing, what does it mean? The authors should be more careful in preparing scientific figures.**

During night-time, the configuration of FMI-PollyXT permitted the determination of the Residual Layer height (RLH). The study of Wang et al. (2016) which was performed at a station of similar latitude, Wuhan, China, revealed that the RLH lies mostly in the range 0.5–1.3 km, following a seasonal variation. Hence, for most of our night-time cases we considered that the lidar system detected the top of the residual layer, which contained the aerosol of the previously mixed layer. In particular, if a layer top more than 500m was detected between sunset and sunrise, it was associated with the RLH.

This definition is now clarified in Section 3.1.1 of the manuscript.

The lidar data was available in 6-hour datasets. For this reason, the algorithm of the WCT method was applied separately to each 6-hour dataset. Furthermore, the color scale of the range-corrected signal contour plots is normalized with respect to the maximum signal recorded in each 6-hour dataset. The 6-hour quicklooks of the lidar-range corrected signal are made available by TROPOS (Leibniz Institute for Tropospheric Research) and can be accessed at http://polly.rsd.tropos.de/?p=lidarzeit&Ort=21.

The width of the black zone in lower part of Figure 3 (top panel) is representing the number of cut-off heights that are used. More specifically, during 00:00-12:00 UTC no cut-off heights were used, 12:00-18:00 UTC 3 cut-off heights (90 m) were used, while during 18:00-00:00 1 height bin was cut off.

These aspects have now been clarified in the manuscript (Label of Figure 3).

**9) P9, L315: PBLH=435 m is not found in Figure 3, check the text, table 2 and Figure 3.**

We thank the reviewer for noticing this mistake. In Figure 3 (lower panel), the PBLH from radiosonde measurements is given correctly according to the  $\theta_{\rm crit}$  and RH\_crit method at 219 m Above Ground Level (AGL). However, in the text (line 315) and in Table 2 we gave the height in meters above mean sea level (ASL) (219 m AGL+ 216 m elevation = 435 m ASL). In the manuscript we kept the PBLH in meters AGL, since the PBLH derived from all of the methods are discussed in meters AGL.As stated above, the WRF-radiosonde profile comparison is not included in the new version of the manuscript.

**10) P9, L333, define PBL cycle, when does this cycle start and end?**

The PBLH cycle is defined as follows:

The PBLH growth period begins when the PBLH started to increase (typically 2-4 h after sunrise) and is complete when 90% of the daily maximum height is reached (typically between 08:00 and 10:30 UTC). More specifically, in the case of 2 March 2009, the PBLH growth period was completed at 7:30 UTC, which was one hour earlier compared to the completion of the PBLH cycle on the previous day. Furthermore, the PBLH started to grow approximately at 4:00 UTC on both days. In the manuscript, the PBL Cycle is defined in Section 3.1.2.

**11) P10, L354: which data did you use to derive this 553m/h? WRF?**

The PBLH growth rate between 3:00 and 5:00 UTC was determined by the cloud base height, which was assumed to be indicative of the PBL top and was derived using the WCT method as described in Section 3.1.2.

**12) Figure 4, and Figure 5: Again, check the color scale please. The discontinuity is quite obvious.**

The lidar data was available in 6-hour datasets. For this reason, the algorithm of the WCT method was applied separately to each 6-hour dataset. Furthermore, the color scale of the range-corrected signal contour plots is normalized with respect to the maximum signal recorded in each 6-hour dataset. The 6-hour quicklooks of the lidar-range corrected signal are made available by TROPOS (Leibniz Institute for Tropospheric Research) and can be accessed at http://polly.rsd.tropos.de/?p=lidarzeit&Ort=21. These aspects have now been clarified in the manuscript (Label of Figure 3).

The colorbars have been removed from Figure 3, 4 and 5, since the colorscale of each 6-hour contour plot is different.

13) P10, Subsection 4.2.2: The data points are so few, and I do not think it is enough for a statistical study. "Based on the analyzed cases, it was found that the overpass distance (here 20 and 101 km) from the lidar station and time difference between the measurements did not affect the PBL heights." This conclusion does not convince me, because the dataset is so small and cannot represent the spatial and temporal variability.

The reviewer is right. Therefore we added the following comment after the statement "based on...PBL heights":

... However, the small number of measurements does not allow us to generalize these findings. Hence, longer measurement periods or more extended comparison to ground stations are needed in order to draw more robust conclusions.

**14) P11: PBLH diurnal Cycle might be more specific than PBL diurnal Cycle, because the authors investigated only the PBL height, not other parameters in the PBL**

In this work, the only parameter that we analyzed was the PBL height (PBLH). For this reason, as the reviewer suggests, we replaced the term PBL with PBLH in the manuscript.

**15) P12, L426: why is the comparison made only between lidar detection and ECMWF, how about WRF?**

The comparison of the PBLH diurnal cycle was performed between lidar and ECMWF Reanalysis because their data availability was sufficient throughout the measurement campaign. On the other hand, WRF simulations were dedicated to specific case studies that are analyzed in Section 4.1 so as to justify the PBLH derived by the WCT method under different aerosol load and meteorological conditions. As mentioned above, in the new manuscript we do not include ECMWF results due to its low horizontal resolution and the need to perform comparison with one atmospheric model in the paper.

**16) P14, Section 5: not relevant and too short to be a section.**

This section has been removed. The comparison with the PBLH characteristics over Elandsfontein site is performed in parallel with the corresponding results (PBLH diurnal and seasonal cycle) from Gual Pahari.

**17) P15 conclusion: this section is long and not conclusive, and it is repeating what have been said previously.**

We thank the reviewer for this comment. After revision of the manuscript content, the Conclusions Section has been rewritten. More emphasis is given on the factors that can explain the patterns of diurnal and seasonal PBLH cycle and contribute to the formulation of PBLH. Moreover, the sources of discrepancies between lidar and numerical estimations are discussed and suggestions for future studies are made. In the same sense, the Abstract has also been revised.

**Planetary boundary layer height by means of lidar and numerical simulationsBoundary Layer variability over New Delhi, India<del>, during EUCAARI project</del>**

1Department of Environmental Physics and Meteorology, Faculty of Physics, University of Athens, Greece 2Alfred Wegener Institute, Helmholtz Centre for Polar and Marine Research, Potsdam, Germany

3Finnish3Finnish Meteorological Institute, Kuopio, Finland

10 Correspondence to: K. Nakoudi (knakoudi@phys.uoa.gr)

Abstract. In this work, the Ground-based lidar measurements were performed at Gual Pahari measurement station, approximately 20 km South of New Delhi, India, from March 2008 to March 2009. The height of the Planetary Boundary Layer (PBLH) is investigated over Gual Pahari, New Delhi, for almost a year. To this end, ground-based measurements from a multi-wavelengthPBL) was

- 15 retrieved with a portable Raman lidar, were used. The system, utilizing the modified Wavelet Covariance Transform (WCT) method was utilized for PBLH retrievals. Results. The lidar derived PBL heights were compared to radiosonde data from Cloud-Aerosol Lidar and Infrared Pathfinder Satellite Observation (CALIPSO) satellite observations and the Weather Research and Forecasting (WRF) model. In order to two atmospheric models. The results were also analyzed on a seasonal basis.
- 20 To examine the difficulties of PBLHPBL lidar detection from lidar, we analyzed three cases of PBLH diurnal evolution under different meteorological and aerosol load conditionsz-we focused on three case studies of PBL diurnal evolution. In the presence of a-multiple aerosol layerslayer structure, the employed algorithmWCT method exhibited high efficiency (r=0.9) in the attribution of PBLH, whereas weak aerosol gradients induced high variability in PBLH. A sensitivity analysis corroborated the
- 25 stability of the utilized methodology. The\_PBL\_height\_determination. Good\_agreement\_with\_the European\_Center\_for\_Medium\_range\_Weather\_Forecasts (ECMWF) and the Weather\_Research\_and Forecasting (WRF) estimations was found (r=0.69 and r=0.74, respectively) for a cumulus convection case. In the aforementioned cases, temperature, relative humidity and potential temperature radiosonde profiles\_were\_well\_compared to the respective\_WRF\_profiles. The Bulk\_Richardson\_Number\_scheme,
- 30 which was applied to radiosonde profile data, was in good agreement with lidar data, especially during daytime (r=0.68). The overall comparison with CALIPSO satellite observations vielded; namely, CALIOP Level 2 Aerosol Layer Product, was very satisfying results (r=0.884), with CALIPSO Feature Detection Algorithms slightly overestimating PBLH. Due to the relatively warmer and drier winter and, correspondingly, colder and rainier pre-monsoon season. PBL height. Lidar measurements revealed that
- 35 the maximum PBL height was reached approximately three hours after the seasonal PBLHsolar noon, whilst the daily evolution of the PBL was completed, on average, one hour earlier. The PBL diurnal cycle during the measurement was also analyzed using ECMWF estimations, which produced a stronger cycle during the winter and pre-monsoon period was slightly weaker. The seasonal analysis of

1

Formatted

<u>KonstantinaK. Nakoudi1,\_2, ElinaE. Giannakaki1,\_3, AggelikiA. Dandou1, MariaM. 5 Tombrou1, MikaM. Komppula3

lidar PBL heights yielded a less pronounced PBL cycle than the cycleone expected from long\_-term climate records. The lowest mean daytime PBL height (695 m) appeared in winter, while the highest mean daytime PBL height (1326 m) was found in the monsoon season as expected. PBL daily growth rates exhibited also a weak seasonal variability.

**1 Introduction**

- The Planetary Boundary Layer (PBL) is the lowermost portion of the troposphere, which experiences a diurnal cycle of temperature, humidity, wind and pollution variations. The PBL height is a key component of the atmosphere and of the climate system, as it fundamentally affects cloud processes, as well as land and ocean surface fluxes. The PBL height (PBLH) is the most adequate parameter to represent the PBL.- Therefore, it is usually required in numerous applications. For-for instance, in pollution-dispersion modellingmodeling, where the upper boundary of the turbulent layer actsplays a
- 50 role as an impenetrable lid for the pollutants emittedreleased at the surface. The PBLHPBL height also appears as a mixing scale height in turbulence closure schemes within climate and weather prediction models (Zilitinkevich and Baklanov, 2001). As air pollution becomes more severe due to economic development, particularly in developing countries (Wang et al., 2009), observations of the PBL height with high temporal and vertical resolution observations of the PBLH are essential for weather and air-
- 55 quality prediction and research. Moreover, the PBLHPBL height is related to the warming rate caused by enhanced greenhouse gases emissions (Pielke et al., 2007). Several methods have been proposed to estimate the PBLHPBL height utilizing vertically resolved thermodynamic variables, turbulence-related parameters and concentrations of tracers (Seibert et al., 2000; Emeis et al., 2004). Different methods for the determination of the PBLH from radiosondes have
- 60 been compared and the associated uncertainties have been estimated (Seidel et al., 2010; Wang and Wang, 2014). Lidar (Light Detection And Ranging)Restrictions of radiosondes refer to the coarse vertical resolution of standard meteorological data with respect to boundary layer studies as well as the smoothing due to the sensor lag constant bounded by the high ascent rate of the radiosonde (Seibert et al., 2000). Remote sensing systems such as aerosol lidar, microwave radiometer (Cimini et al., 2013).
- 65 wind-profiling radar (Cohn and Angevine, 2000) and Doppler wind lidar (de Arruda Moreira et al., 2018) are suitable for long-termean provide continuous measurements of various atmospheric quantities with high temporal resolution and can be used either independently or synergistically to retrieve the PBLH. Space-borne lidar systems provide the advantage of spatial coverage, although for studies focusing on a particular area of interest, measurements are constrained by the overpass
- 70 frequency. Ceilometers are simple backscatter lidars, which entail less operational cost. However, exploitation of their full potential is on an early stage with limited ceilometer-related studies (Münkel et al., 2007, Binietoglou et al., 2011, Wiegner, including the vertical distribution of et al., 2014). Ceilometers have high potential of contributing to PBLH climatology, within certain limits, but detailed investigation of open issues is still needed, as for example, the treatment of incomplete
- 75 overlap. Additionally, no adjustments can be typically made by the user, contrary to the modified Wavelet Covariance Transform (WCT) algorithm. Hence, improvements on layer detection algorithms are urgently needed to fully exploit the potential of ceilometers. In elastic and Raman lidar systems,

the atmospheric aerosols are used as tracers and the PBLH is indicated by a gradient in the rangecorrected lidar signalfrom which the PBL height can also be retrieved (Menut et al., 1999; Cohn and

- 80 Angevine, 2000; Brooks 2003; Amiridis et al., 2007; Morille et al., 2007; Baars et al., 2008; Engelmann et al., 2008; Groß et al., 2011; Tsaknakis et al., 2011; Haeffelin et al., 2012; Cimini et al., 2013; Soria et al., 2013; Korhonen et al., 2014; Lange et al., 2014; Bravo-Aranda et al., 2016). Weather and climate prediction models could alternatively be used to determine the PBLH. especially for strong horizontal inhomogeneity. However, inconsistencies in the definition
- 85 of PBLH among the existing meteorological models also result in significant differences in its calculation (Tombrou et al., 2007).; de Arruda Moreira et al., 2018). Atmospherie aerosols are used as tracers and the PBL top is indicated by a gradient in the range corrected lidar signal. New Delhi is one of the most densely populated cities, with 29259 inhabitants per square mile, and the

[revised manuscript text omitted]

- 185 Tropospheric Research) and can be accessed at http://polly.rsd.tropos.de/?p=lidarzeit&Ort=21.(2008). http://polly.rsd.tropos.de/?p=lidarzeit&Ort=21.(2008). The WCT method makes use of the assumption assumption that the PBL contains much more aerosol load compared to the free troposphere and, thus, troposphere and, thus, a strong backscatter signal decrease can be considered asobserved at the The covariance transform  $W_t(a,b)$  is based ona measure of the convolution ofsimilarity between the
- 190 signal and the related Haar function (Baars et al., 2008). This method was chosen because it allows allows larger adjustability than other techniques, as shown from previous studies (Baars et al., 2008; al., 2008; Korhnonen et al., 20132014). For instance, the gradient technique involves an ambiguity in

Adjust space between Asian text and numbers

| Formatted: English (United States) |  |
|------------------------------------|--|
| Formatted: English (United States) |  |

ambiguity in the choice of the "relevant" minimum in the gradient that corresponds to the PBLH PBLH boundary layer height (Lammert and Bösenberg, 2005).

- 195 the WCT threshold, threshold value of the WCT which allowedpermits the identification of a corresponding omission of weak gradients, was introduced as a first modification. The first height maximum of Wf(a,b) occurred, exceeding the selected signal decrease threshold, was defined as the defined as the PBLH. A second modification introduced by Baars et al. (2008) was related to strong gradients PBL height. This threshold was modified in eases of multiple aerosol layers structures, where
- 200 strong gradients inside the PBL complicated the detection of the PBL height. Furthermore, the option to cut the lower parts of the PBL (30-870 m) and the ability to exclude these parts from the lidar data evaluation. In this work, the signal (from 30 to 870 m) was utilized so as to avoid strong gradients related to the incomplete overlap in the lower heights. The importance of a proper threshold adjustment is discussed in Section 4.1, where three case studies are analyzed and the applicability of
- 205 the WCT techniquemethod, under different meteorological and aerosol load conditions\_is\_discussed (Section 4.1) in the context of three case studies and the stability of the WCT algorithm, is assessed as well (Section 4.2). Additional cases, where the importance of a proper threshold and cutting-off zone are discussed, can be found in Nakoudi et al. (2018). examined.

Daily mean and maximum PBLH correspondsPBL heights correspond to convective hours (3:00-12:00
 UTC). The hourly PBLH wasPBL height values were calculated from the 15\_min lidar observationsdate by averaging of the three closest data points of the time considered (e.g. 12:00 hourly height would be the average of the three data points between 11:45 and 12:15). The seasonal cycle study was based on the classification proposed by the Indian Meteorological Department, i.e. winter (December-March), pre-monsoon or summer (April-June), monsoon (July-September) and post-

215 monsoon (October-November) (Perrino et al., 2011). However, the PBLHPBL seasonal cycle was examined during the winter, pre-monsoon and monsoon periods, as no sufficient data coverage was found during the post-monsoon period(Section. The diurnal PBL cycle is provided by lidar measurements and ECMWF estimations for the whole measurement period as well as on a seasonal basis (Section 4, 3, 1, 3).

**220 3.1.3 Data coverage**

225

During the one\_-year long measurement campaign, from 12 March 2008 to 31 March 2009, FMI-PollyXT was measuring on 139 days. Due to technical problems with the laser, (27%), the data coverage from September to January was sparse. Furthermore, precipitation prohibited lidar measurements, since the lidar system had to shut down. Hence, sufficient (12%). Thus, lidar measurements were possible in 61% of the total time (139 days).

Sufficient data availability (more than 25%, from 4 h after sunrise to 1 h before sunset)-was achieved during 72 days. MultipleDuring these days, multiple aerosol layers appeared mainly between March layer structures (20%) and May, whereas low clouds were present mostly in the monsoon period and both(15%) complicated PBLHPBL height detection. Additionally, some technical issues arose due to

230 photomultiplier supersaturation and signal problems. (9%). A lack of a significant decrease in the backscatter profile was observed in only a few cases. (3%). The latter was a first indication that the

| Formatted: English (United States) |
|------------------------------------|
| Formatted: English (United States) |
| Formatted: English (United States) |
| Formatted: English (United States) |

modified WCT method cancould detect the PBLHPBL top efficiently, as long as the signal decrease threshold was tuned properly. TheHence, the PBL height could be identified in 53% of the cases with sufficient data availability (72 days). In Figure 2, the data coverage is presented on a monthly basis in

Figure 2, The highest PBLHPBL detection frequency was achieved in February, whichreaching 74%. This high detection rate can be attributed to favorable meteorological conditions, since in February the occurrence of low clouds appeared sparsely without anywas 0.7% with no rainfall events.

**3.2 Radiosonde measurements**

235

240

245

The Bulk Richardson Number (BRN) method was used for PBL determination, employing the formula introduced by Menut et al. (1999):

$$Ri_{b}(h) = \frac{g(h - h_{0})}{\theta(h)} \frac{[\theta(h) - \theta(h_{0})]}{u(h)^{2} + v(h)^{2}}$$

, where h is altitude,  $h_0$  the altitude of the ground, g gravitational acceleration,  $\theta$  potential temperature in Kelvin and u and v the zonal and meridional wind components, respectively. The PBL height was determined to be the lowest altitude where BRN reached the critical value, which is taken equal to 0.21 (Vogelezang 1996). Beyond this critical value of Ri, the atmosphere can be considered stable and fully

decoupled from the underneath layer.

**3.23 Space-borne lidar observations**

[revised manuscript text omitted]

stable

<del>similar</del>

- 315 capabilitiestheir strengths and limitationslimits. First, the evolution of PBLHPBL under cloudless conditions is discussed for 12 February 2009. Subsequently, a two-day case with a multiple aerosol layer structure is presented for 1-2 March 2009. Finally, the diurnal development of PBLHPBL is investigated in the presence of low clouds for 29 June 2008, The three criteria (T\_crit, RH\_crit, 0\_crit) were used to determine PBL height in each radiosonde profile. These criteria were also applied to WRF
- 320 It was found that the presence of multiple aerosol layers and low clouds can pose difficulties in PBL top detection (Section 3.1.3). However, as it will be shown these difficulties can be dealt with the use of proper WCT threshold and cut off values (see Section 3.1.2).
- The diurnal evolution of PBL during 12 February 2009 was characterized by an almost constant daily growth rate (133 m/h between 06:00 UTC and10:00 UTC) with a maximum height of 950 m (is presented in Figure 3).-Sunrise was approximately at 01:30 UTC, while sunset was at 12:40 UTC. No aerosol layers were observed in the free troposphere. Althoughfound aloft. Between 06:00 12:00 UTC,
- although internal gradients (yellow and red color) of aerosol content, appeared inside the PBL (06:00-12:00 UTC),5 the default signal decrease threshold (0.05) was efficient. However, later (Between 12:00-18:00 UTC) in order to avoid strong gradients in the lower parts of the PBL, higher -a-threshold
   (of-0.08) was used in conjunction with a 90 m cut-off heights (90 m), Furthermore, height, Due to low
- 330 (of-0.08) was used in conjunction with a 90 m-cut-off heights (90 m). Furthermore, height. Due to low aerosol load conditions were responsible for high variability incontent, the derived PBLH (PBL heights between-12:00-14:00 UTC)\_-showed high variability. An almost constant daily growth rate of 133 m/h was found from 06:00 UTC to 10:00 UTC. The maximum height of 950 m was reached at 10:30 UTC. During convective hours (05:00-12:00 UTC), WRF overestimated the PBLH mainly due to the
- 335 simulated neutral profile-virtual potential temperature at the surface similar to that around 1100 m AGL (differences < 0.5 K, not presented), resulting in an increase in the PBLH (Kim et al., 2013). It is worth mentioning that during the convective period FMI-PollyXT identified a light aerosol load activity at the altitude where the numerical models estimated the PBLH, with the WCT technique not detecting this activity due to the weakness of the aerosol gradients. During night-time model estimations yielded
- 340 lower PBLH compared to lidar data. During night-time model estimations yielded lower PBLH compared to lidar data. The low wind field produced by the WRF close to the surface (wind speed values up to 3 m/s in the first kilometer) and, thus, the lack of sufficient mechanical turbulence, can be related to the shallow nocturnal PBL. It should be noted that the measured PBLH is expected to depict, apart from any mechanically-driven layer during the stable and transition periods, the top of the 345 previous day's residual aerosol layer, while the simulated PBLH from WRF refers to the height of the

shallow mixed layer. Therefore, their difference is expected since they depict different layers. The overall correlation was satisfying (r=0.8).

Figure 3 (middle panel) shows the development of PBL according to FMI PollyXT measurements and the estimations from the two atmospheric models. During convective hours (05:00-12:00 UTC), the
 WRF and ECMWF models seemed to overestimate PBL height. On the other hand, during night time model estimations yielded lower PBL heights compared to lidar data, since the former estimated the nocturnal PBL while the latter identified the RL top. Between 6:00 and 12:00 UTC, FMI PollyXT identified a light aerosol load activity at the altitude where the WRF model estimated the PBL height. The correlation with lidar hourly heights was satisfying (r=0.8) for both model output data, while a

- 355 During the two\_day period of 1-2 March 2009, a complex aerosol layer structure appeared in the free troposphere up to 3 km altitude (Figure 4). However, appropriate modification of the modified WCT method managed to detect the top of the PBL in most of the 15 min intervals, after modifying the signal decrease threshold and use of appropriate applying a cut-off heights allowed for the detection of PBLH. In order to avoid gradients in the lower parts of the PBL, the signalheight. The threshold was adjusted
- 360 (within the range of 0.03-0.08) within, which corresponds to a 6-16% signal decrease, in combination withrespectively. Furthermore, a 30-60 m cut-off zoneheight was used, in order to avoid gradients in the lower parts of the PBL.

On 1 March 2009, the transition period (02:00<del>\_to</del>-05:00 UTC) was characterized by a slow PBLHPBL development (of-14 m/h), whereas the PBLHPBL evolution was more pronounced in the convective period (05:00-to-09:00 UTC) with a mean growth rate of 101 m/h. The maximum height (of-950 m)

- appeared at 08:45 UTC. On the next day, a stronger but slightly shorter PBLHPBL cycle was observed, with a mean evolution rate of 187 m/h, reaching a maximum height (of-1010 m) at 08:15 UTC15UTC. This slight modification in PBLHthe development of the PBL, can be attributed to the combination of higher temperature and lower wind speed conditions duringcharacterizing the second day.
- 370 4.1.3 Case with low clouds: 29 June 2008

375

In this case broken cumulus clouds appeared between 600-1100 m (from 00:00 to 12:00 UTC). On average, a moderate PBLH evolution (86 m/h) was found, with a maximum height (1279 m) appearing at 9:15 UTC (Figure 5). Whenever clouds appeared below 1km, we made the. The assumption that the cloud base isconstitutes an approach to the top approximation of the PBL, however top was made. However, it could be argued that the PBLHPBL top was located at a higher level, wheresince diffuse

aerosol layers were found. In additionalso present there. During this period, it was difficult to findlocate an adequate signal decrease; the default \_gradient; a threshold <del>corresponding to 10%</del> <del>decrease</del>-was used, while sensitivity tests with thresholds sensitive to weaker gradients<del>lower threshold</del> <del>values</del> yielded the same results. Hence, the algorithm exhibited -decreased sensitivity, which can be attributed is mainly related to the existence of diffuse aerosol layers. High PBLH was observed

immediately after, due to Large PBL height values also appeared around 12:00 UTC, corresponding to a strong aerosol layer which sprawled to lower heights, either through probably due to dry removal or precipitation that Formatted: English (United Kingdom)

evaporated before reaching the ground. Following a short Rainfall was observed between 13:30 and
 remaining aerosolaerosols kept being displaced downwardsin the downward direction, creating strong the The effect of aerosol removal effect was clear (can be seen between 16:00- and 24:00 UTC) aerosol load, observed, complicated the detection of PBLH was complicated and, hence, accounted the PBLH (the detected PBL heights between 16:00- and 24:00 UTC).

- WRF and ECMWF estimations correlated well with FMI-PollyXT hourly PBL height data (r=0.74), and
   r=0.69, respectively). During daytime, WRF slightly overestimated PBLHPBL height, while it should be noted that FMI-PollyXT identified intermittent aerosol gradients at an underestimation was observed during night-time by both models. Good agreement was corroborated by additional statistical parameters. Fractional bias was equal to 0.015 and 0.11 for WRF and ECMWF estimations, respectively.
- 395

**4.2 Comparison of lidar PBL heights to ancillary data sources**

During the measurement period, 24 CALIPSO overpasses were available withininside 1° radius around Gual Pahari station. The Boundary Top LocationIn 17 cases the boundary top location algorithm (SIBYL (-Selective Iterated Boundary Locator) identified two to four layers (17 cases), while, whilst in the remaining7 cases no layers were identified. However, For the 17 cases, the PBL top from the ground-based lidar observations were not was-available in all for 14 cases. In one case, the top of the cases (only in 14).second layer was chosen, as the first one was inside the PBL, according to the attenuated backscatter image from CALIOP. Furthermore, some5 cases (5) were excluded fromnot included in the comparison as the detected layers were clearlyeither above the typical\_PBL limits (higher than (height >3 km).) or in the free troposphere (height >10 km).

**4.3 Statistical Analysis**

**4.3.1 PBL Diurnal Cycle of PBLH**

Figure 8a shows the mean diurnal PBL evolution as obtained by lidar measurements and ECMWF estimations. Although night-time PBLH isPBL height values were not taken into account for the statistical analysis of PBL-seasonal PBLH (Section 4.4.2), height, nocturnal PBLH is considered values are included here in orderso as to investigatepresent the PBL-diurnal evolution.
 ECMWF estimations revealed a shorter but stronger PBL growth period, with a maximum top height of 2137 ± 143 m, which appeared earlier than the one given by FMI-PollyXT. As in the annual and winter

diurnal cycle, ECMWF overestimated PBL top height during convective hours. On the other hand,
 underestimation was observed during the early morning hours, with a more significant underestimation during night-time due to the fact that FMI PollyXT identified the RL, whereas the ECMWF estimated the nocturnal PBL top. The total comparison reached an r of 0.84.

In this Sectionthe following, we statistically analyzepresent the main statistical findings regarding the lidar measurements in conjunction with the during 72 days. The seasonal cycle of mean and maximum seasonal mean PBLHPBL height was found at 695 ± 146 m during winter(17 days), 878 ± 297 m

period (15 days) and 1025 ± 296 m during the monsoon. The (40 days). Regarding the seasonal

at  $1191 \pm 516$  m during winter,  $1326 \pm 565$  m during the pre-monsoon period and at  $1361 \pm 350$  m during the monsoon. In general, the PBLH seasonal cycle followed the temperature cycle very well. The temperature cycle of the During the measurement days was fairly representative of the whole

- 425 seasonal cycle, with the temperature distribution being similar to the distributions of the whole seasonal periods. During the measuring period, a mean temperature of  $21 \pm 4$  °C was found in the winter,  $27 \pm 3$  in the pre-monsoon and  $30 \pm 2$  °C in the monsoon season while the. A seasonal average maximum temperature of  $29 \pm 5$  °C was recorded at  $29 \pm 5$  °C in the winter,  $33 \pm 4$  °C in the pre-monsoon and  $35 \pm 2$  °C accordingly. Nevertheless, it should be mentioned that the monsoon period.
- 430 In winter, the daily mean PBLH distribution was narrower (in majority between 600 and 900 m) compared to the pre-monsoon and monsoon seasons (mostly between 900 and 1200 m). Following a similar pattern, the daily maximum PBLH was rather confined in winter (in majority between 900 and 1200 m) with a significantly broader spectrum (between 600 and 1800 m) in pre-monsoon and monsoon. The highest inter\_seasonal variability was exhibited during the-pre-monsoon-season both in
- 435 terms of mean and maximum PBL height, which couldmay be attributed to the meteorological conditions. The of this period. During the pre-monsoon season comprised days, 7 eases with heavy rainfall and days(7-37 mm daily accumulated precipitation) and 8 cases with hardly any precipitation, which can potentially explain the appeared (less than 3 mm accumulated precipitation). This combination led to a broad distribution of daily mean PBLH (PBL heights (from 251-m to-1191 m). In
- 440 winter large Large inter\_seasonal variability was also observed in the winter period, in terms of attributed to the broad inter\_seasonal range of maximum temperature range, which was almost  $16 \, ^{\circ}\text{C}$  (20  $^{\circ}\text{C}$  36  $^{\circ}\text{C}$ ).

The frequency distribution of daily mean PBL height is presented in Figure 9 for 6 different classes of 300 m. During the measurement campaign, the majority of daily mean PBL heights were found between the classes of 600 and 1200 m (40% within 600 900 m; 32% within 900 1200 m). The winter period distribution was narrower and skewed towards the 600 900 m class. In the pre-monsoon and monsoon seasons, PBL height distributions were quite broader with a maximum between 900 and 1200 m. In terms of daily maximum PBL height, the majority of heights were found between 900 and 1800 m (26% within 900-1200 m; 22% within 1200-1500 m; 29% within 1500-1800 m). In the winter

450 period, a more confined distribution appeared, with 53% of daily maximum heights between 900 and 1200 m. The PBL height spectrum was significantly broader in the pre-monsoon and monsoon periods, with maximum daily heights to spread between 600 and 1800 m.

During the The distribution of daily growth rates is presented in Figure 9. For the whole, measurement period, daily evolution rates were mostly within observed in the 100-200 m/h but lower rates (elass,

- while a significant number of mean growth rates was observed between 29-100 m/h). Different frequency distributions were observed as well. In winteron each seasonal period, albeit the average evolution rates did not exhibit strong seasonal variability. In the winter period, daily growth rates presented a slightly broad distribution (mostly with most of them lying between 100 and 200 m/h) with (40%), while a mean evolution rate of 157 ± 81 m/h (Figure 8).N=15) was found. In the pre-monsoon, slightly-season, higher growth rates were observed (mainly within 100-300 m/h), with an average of 3
- $206 \pm 134$  m/h. Additionally, rates between 0-100 m/h and 500-600 m/h (N=9), with 44% of them

| 1                | Formatted: English (United States) |
|------------------|------------------------------------|
| 1                | Formatted: English (United States) |
| 1                | Formatted: English (United States) |
| 1                | Formatted: English (United States) |
| /                | Formatted: English (United States) |
| 1                | Formatted: English (United States) |
| λ                | Formatted: English (United States) |
| 1                | Formatted: English (United States) |
| -(               | Formatted: English (United States) |
| 1                | Formatted: English (United States) |
| $\left( \right)$ | Formatted: English (United States) |
| -                | Formatted: English (United States) |
|                  |                                    |

within the range 100 200 m/h, while 33% were observed, following the pattern of between 200 and 300 pre-monsoon season (Section 4.4.2). In the monsoon season, lower evolution speeds were slightly lower observed (121  $\pm$  67 m/h). N=22), with 45% being less than 100 m/h., while a significant percentage (40%) was found between 100 and 200 m/h.

The average PBL height was lower in Gual Pahari (866 m ) in comparison to Elandsfonteiner (1400 m) with less seasonal variability (standard deviation of 165 m in Gual Pahari; 500 m in Elandsfontein). In both sites the maximum PBL height was reached approximately three hours after the solar noon, since the daily solar cycle is similar in the latitudes of the two

470 stations. In Gual Pahari, the highest rates (mostly within 100-300 m/h) appeared in the premonsoon-season (April-May), whilst in Elandsfontein maximum rates (between 120-320 m/h) were reached during spring\_-{September-October, (Kohronen et al., 2014) a period that exhibits strong similarities with the }. The pre-monsoon season in India. - and the spring season in South Africa have strong similarities.

**475 56 Summary and Conclusions**

465

In this studypaper, one year long ground-based lidar measurements were used to retrieve PBLHanalyze PBL height variability over Gual Pahari, New Delhi. The feasibility of deriving PBLH with the modified WCT technique was investigated and the respective resultslidar retrieved PBL heights were compared to data from independent sources.

- 480 In: radiosondes, satellite observations and two atmospheric models. Three case studies of PBL daily evolution were discussed so as to identify atmospheric structures which can complicate PBL height detection. It was found, in support of previous work (Baars et al., 2008; Korhonen et al., 2014), it was found2013), that the modified WCT method exhibited satisfying efficiency performed well-under different meteorological and aerosol load regimes. On a case with elevated aerosol layers, More
- 485 specifically, a significantly good performance was revealed, even when the layers were injected into the PBL. Such layers have been reported in literature as-on a major challenge in the attribution of the PBLH especially during night-time (Haeffelin et al., 2012). PBLHtwo day case, with multiple aerosol layers aloft. However, PBL determination was complicated in the presence of before a rain event, where lofted layers created strong aerosol content gradients and later on, where diffuse aerosol layers.
- 490 Low aerosol load, observed mainly during morning or afternoon transitions, also represents a condition for uncertain determination of PBLH (Haeffelin et al., 2012). Sensitivity analysis revealed stable performance of the WCT algorithm, with the exception of elevated layers and PBL internal gradients, which affected the results when specific thresholds were applied. Higher thresholds appeared to be more sensitive towards detecting lofted layers.
- 495 In the context of the aforementioned cases, WRF model case studies, numerical estimations overestimated PBLHPBL height in the daytime, while an underestimation was observed in the night-time. The understanding of turbulence in nocturnal SBL and its parameterization is rather slow and not well established in NWP models (Mahrt et al., 1999; Beare et al., 2006; Hong 2010). In this study, this

| Formatted: English (United States) |
|------------------------------------|
| Formatted: English (United States) |
| Formatted: English (United States) |

| Formatted:
States) | Font: | Times Nev | v Roman, | 10 pt, | English (United |
|-----------------------|-------|-----------|----------|--------|-----------------|
| Formatted:
States) | Font: | Times Nev | v Roman, | 10 pt, | English (United |
| Formatted:
States) | Font: | Times Nev | v Roman, | 10 pt, | English (United |

is The latter can be partly addressed by the revised SBL scheme that retains the turbulent levels so as to 500 avoid the abrupt collapse of the mixed layer after the sunset by using the exchange coefficients.attributed to the fact that the lidar system identified the RL, whereas the numerical models estimated the nocturnal PBL top. The comparison between radiosonde and WRF vertical profiles, through three different methods, showed that radiosonde data overestimated PBL height in the night time. The discrepancies between radiosonde and WRF PBL heights could be attributed to various 505 sources, such as the different vertical resolution and the different nature of each data set; radiosondes provide in situ measurements, whereas WRF model provides numerical estimations of various meteorological parameters. However, the fact that neither anthropogenic heat sources nor heat storage in buildings wereare included in the simulations could also explain the model underestimation. Furthermore, it should be noted that the measurements often depict different layers from the simulated 510 ones, as in the case of the residual aerosol layer. During the rainy season of monsoon, the diurnal cycle of PBLH was weaker and its evolution was completed earlier. A relatively warmer and drier winter and, respectively, a colder and rainier premonsoon were observed compared to climatological records. These meteorological patterns could account for the observed PBLH cycle, which was rather indistinct compared to the cycle expected from 515 long-term climate statistics. Daily evolution rates of 29-200 m/h were mainly observed, with lower rates during the rainy season of monsoon. The evolution of PBL started two to four hours after sunrise and was completed two hours after the solar noon, with the maximum PBL height observed approximately one hour later. In the winter and pre-monsoon season, ECMWF data revealed a stronger PBL daily evolution. During the monsoon season, both FMI-PollyXT measurements and ECMWF output data, produced a smoother diurnal cycle, consisting of weaker fluctuations between daytime and 520 night\_time, with PBL heights from ECMWF being systematically lower than those derived from FMI-Future studies are necessary in order to better understand the factors that modulate the exchange of moisture, heat and momentum between the surface and PBL and, consequently, affect the comparison of modelled PBLH with observational data. In addition, the relative contribution of the various PBL 525 dynamics drivers, under different aerosol load and meteorological regimes, needs to be further investigated. The feasibility of applying the modified WCT method in simpler lidar systems such as ceilometer and Doppler lidar, should be assessed. These systems entail less operational cost and, thus, exhibit good potential for determining the PBLH and evaluating weather prediction and pollution dispersion models on an operational basis. In recent years, significant effort has been made towards the 530 establishment of ceilometer networks by national weather services and other agencies over Europe with the aim to build up a framework for real-time applications and improvements of air quality and weather prediction by assimilation of ceilometer data (Haeffelin et al., 2012; Wiegner et al., 2014). Analogous efforts are currently in progress over different parts of India, like in the states of Maharashtra and Kerala and in the union territory of Delhi (Sharma et al., 2016; Babu et al., 2017; 535 https://www.lufft.com/projects/several-lufft-chm-15k-ceilometer-projects-in-india-529/). The seasonal PBL cycle observed during the measurement campaign was less pronounced than the one expected from climatological records. This could be attributed to the combination of a relatively warmer winter

and a colder pre monsoon period with respect to long term climate statistics. The highest values of

[revised manuscript text omitted]